# Biomarker-Based Precision Therapy for Alzheimer’s Disease: Multidimensional Evidence Leading a New Breakthrough in Personalized Medicine

**DOI:** 10.3390/jcm13164661

**Published:** 2024-08-08

**Authors:** Anastasia Bougea, Philippos Gourzis

**Affiliations:** 11st Department of Neurology, National and Kapodistrian University of Athens, 15772 Athens, Greece; 21st Department of Psychiatry, University of Patras, 26504 Rio, Greece; pgourzis@upatras.gr

**Keywords:** Alzheimer’s disease (AD), precision medicine, biomarkers, genome-wide association studies (GWAS)

## Abstract

(1) Background: Alzheimer’s disease (AD) is a worldwide neurodegenerative disorder characterized by the buildup of abnormal proteins in the central nervous system and cognitive decline. Since no radical therapy exists, only symptomatic treatments alleviate symptoms temporarily. In this review, we will explore the latest advancements in precision medicine and biomarkers for AD, including their potential to revolutionize the way we diagnose and treat this devastating condition. (2) Methods: A literature search was performed combining the following Medical Subject Heading (MeSH) terms on PubMed: “Alzheimer’s disease”, “biomarkers”, “APOE”, “APP”, “GWAS”, “cerebrospinal fluid”, “polygenic risk score”, “Aβ42”, “τP-181”, “ p-tau217”, “ptau231”, “proteomics”, “total tau protein”, and “precision medicine” using Boolean operators. (3) Results: Genome-wide association studies (GWAS) have identified numerous genetic variants associated with AD risk, while a transcriptomic analysis has revealed dysregulated gene expression patterns in the brains of individuals with AD. The proteomic and metabolomic profiling of biological fluids, such as blood, urine, and CSF, and neuroimaging biomarkers have also yielded potential biomarkers of AD that could be used for the early diagnosis and monitoring of disease progression. (4) Conclusion: By leveraging a combination of the above biomarkers, novel ultrasensitive immunoassays, mass spectrometry methods, and metabolomics, researchers are making significant strides towards personalized healthcare for individuals with AD.

## 1. Introduction

In 2011, the National Institute on Aging and the Alzheimer’s Association (NIA-AA) suggested that the clinical staging of Alzheimer’s disease (AD) ranged from the preclinical stage to the mild cognitive impairment (MCI) stage or the dementia stage [1,2]. Just 19% of MCI patients do not have any neurodegenerative pathology, whereas 30% have non-AD pathology and 51% of MCI individuals show signs of amyloid pathology [3]. In the presence of atypical presentations, such as primary progressive aphasia (PPA) of logopenic type [4], in the early disease, in the community, and in the existence of comorbidities, it is long known that clinical diagnostic accuracy may drop substantially [5] and that up to 39% of patients in whom a non-AD diagnosis was given during life will prove to have AD at autopsy [6]. The opposite is also true, and up to 30% of patients diagnosed with AD will prove to have a non-AD pathology at neuropathological examination [7]. Thus, the in vivo clinical diagnosis of AD is probabilistic, and postmortem verification (or ruling out) remains the gold standard for final diagnosis.

A biomarker is defined as “a characteristic that can be objectively calculated and evaluated as a measure of physiological biological course, pathogenic processes, or pharmacological responses to a therapeutic intervention” [8]. Considering that (a) obtaining in vivo brain tissue samples is a highly invasive method and (b) several CNS-related processes are mirrored in the cerebrospinal fluid (CSF) and amyloid/or tau positron emission tomography (PET), the latter could be an ideal source of biomarkers for detecting and monitoring various pathophysiological processes [9]. Recently, the revised criteria proposed the biological diagnosis of AD based on CSF or plasma and imaging biomarkers that are subclassified according the proteinopathy or pathophysiological pathway (A, T1, T2, N, I, V, S) [10]. Core 1 biomarkers, such as α-amyloid Aβ42, phosphorylated tau (pTau 181), (p-tau217), (p-tau231), and amyloid PET, determine the early AD phage that is detectable in vivo and can identify the presence of AD in both symptomatic and asymptomatic individuals [10]. Considering their time of onset, plasma p-tau217 and p-tau231 have been suggested as biomarkers of Aβ plaques, but this is not conceptually correct because of the coexistence of tau fragments. Core 2 biomarkers (MTBR-tau243, p-tau205 non-phosphorylated mid-region tau fragments, and Tau PET) become abnormal later in the evolution of AD and inform on the risk of short-term progression in people without symptoms. The biomarkers p-tau217, p-tau181, and p-tau 231 were demonstrated to augment at the beginning of Aβ aggregation prior to the modification in tau-PET, and p-tau205 and t-tau started to elevate near the outbreak of clinical symptoms [11]. Currently, the following biomarkers have sufficient accuracy to be diagnostic of AD: amyloid PET; CSF Aβ 42/40; CSF p-tau 181/Aβ 42, CSF t-tau/Aβ 42; or “accurate” plasma assays, where “accurate” is interpreted as accuracy that is equivalent to endorsed CSF assays in discovering abnormal amyloid PET in the intended-use population [12].

According the International Working Group on Mild Cognitive Impairment (IWGMCI), different cognitive phenotypes could arise from different cognitive domains being affected independently of memory and the fact that subjective complaints were no longer necessary [13]. These new criteria, which provide etiological and prognostic characterizations of clinical utility, include the distinction between amnestic and non-amnestic MCI subtypes as well as whether cognitive impairment is restricted to a single domain or numerous domains. The IWGMCI agreement said that biomarkers could be useful in clarifying clinical progression and offered a flexible framework for MCI diagnosis. Aβ1–42 and Tau together showed up to 95% sensitivity and 83% specificity in identifying MCI patients who progressed to AD [14]. Nevertheless, [11C] PIB PET imaging may be able to distinguish prodromal AD patients more accurately than CSF biomarkers [15]. It would be beneficial to use therapy in the early stages of the disease, when these interventions may be more successful, in order to anticipate the progression of these MCI patients towards dementia.

According to the National Institute of Health (NIH), precision medicine, otherwise named personalized medicine, purposes to adjust medical interventions to the individual characteristics of each AD patient, including their environmental, lifestyle, and genetic makeup [16]. Variations in the Apolipoprotein E APOE gene, particularly the APOE4 allele, are a well-studied risk of developing AD because they are involved in decreased β-amyloid clearance, elevated microglial proinflammatory activation, disturbed glucose and lipid metabolism, and synaptic disorganization [17]. Medical factors participate in the manifestation of AD, such as preexisting comorbidities such as cerebrovascular disorders, diabetes, hypertension, epigenetics, and inflammation [18]. In the context of AD, precision medicine holds great promise for improving early detection, prognosis, and treatment outcomes by leveraging individual risk factors to guide clinical decision-making.

In this narrative review, we will explore the latest research on genetic, fluid (CSF and blood), and neuroimaging biomarkers of precision medicine for AD and their potential applications in personalized healthcare.

## 2. Materials and Methods

Even though the aim of this review is not to conduct a systematic review, we employed the basic principles of a systematic review, limiting it to published peer-reviewed articles and a narrative analysis [19]. A literature search was performed combining the following Medical Subject Heading (MeSH) terms on PubMed: “Alzheimer’s disease”, “biomarkers”, “APOE”, “APP”, “GWAS”, “cerebrospinal fluid”, “polygenic risk score”, “Aβ42”, “p-181”, “ p-tau217”, “ptau231”, “proteomics”, “microRNA”, “total tau protein”, and “precision medicine” using Boolean operators. The snowballing procedure was carried out to screen the references of each selected article for potential extra papers to cover the current key evidence.

### 2.1. Inclusion Criteria

The inclusion criteria were relevant in vivo and vitro studies published in English, including society recommendations, international consensus and practice guidelines, and expert panel reports published through May 2024.

### 2.2. Exclusion Criteria

(1) Dementia syndromes apart from AD; (2) reviews, letters, editorials, conference papers, and theses; and (3) papers that did not present results were ruled out.

## 3. Results

According to the flowchart of this review, we eliminated 1525 duplicates from the initial screening of 1923 studies. We revised 221 articles that satisfied the title and abstract of the inclusion criteria. Lastly, subsequent to a full-text review, 118 were chosen for the narrative analysis (Figure 1).

Biomarkers for the early identification of AD have been categorized into five main groups: biochemical, neuroanatomical, metabolic, neuropsychological, and genetic. The biochemical group include cerebrospinal fluid (CSF) and blood-based (plasma/serum, platelets, and peripheral blood mononuclear cells) biomarkers [20,21]. The neuroanatomical group contains computed tomography (CT) and magnetic resonance imaging (MRI) scan biomarkers, while in the metabolic category, there are positron emission tomography (PET) scan and single photon emission computed tomography (SPECT) scan biomarkers [22]. Genetic biomarkers incorporate mutations in the amyloid precursor protein (APP) and presenilin genes (PSEN1 and PSEN2) [23] that are responsible for early-onset AD as well as a major genetic risk factor for late-onset AD, the apolipoprotein E gene (APOE). The APOE genotype is an inherent risk marker rather than a biomarker for Aβ pathology (the CSF Aβ tests identify cerebral Aβ pathology but not the APOE genotype). Awareness of the APOE genotype has, however, earned enhanced clinical importance in the framework of anti-Aβ immunotherapy. A recent study by Fortea et al. [24] suggested that APOE4 homozygotes represent a genetic form of AD with characteristics such as approximately in-depth penetrance, the likelihood of symptom outbreak, and the foreseeable sequence of biomarker changes. The risk of amyloid-related imaging abnormalities (ARIAs) is significantly higher in APOE ε4 homozygotes than in heterozygotes and non-carriers. Therefore, the FDA label for lecanemab consists of screening for APOE and counseling for homozygotes. APOE4 status should be accepted as a crucial parameter in clinical trial design, patient retrieval, and data evaluation, with AD risk across age, sex, race, and ethnicity (stronger risk for East Asians vs. Hispanics) for establishing personalized AD therapy [25].

### 3.1. Classical Neurodegenerative Biomarkers

During the last decade, the three “established” or “classical” cerebrospinal fluid (CSF) biomarkers for AD have been incorporated in diagnostic criteria/guidelines [1,4] and a classification system (ATN) [26]. The ATN research scheme, suggested in 2011 and updated in 2018 by the NIA-AA, recommended the application of biomarkers (namely amyloid (A), tau (T), and neurodegeneration (N)) to diagnose individuals with AD. This classification was conceived for a biological, not a clinical, diagnosis of AD. This ATN research context employs CSF biomarkers where (a) the ratio of the two amyloid-β Aβ peptides (CSF Aβ42/40) is an estimation for A amyloid-β peptide with 42 amino acids (Aβ42), which is decreased in AD, and is considered a marker of amyloid plaque pathology [27]; (b) tau phosphorylated at threonine 181 (p-Tau 181) is an estimation for T tau protein phosphorylated to a threonine residue at position 181 (τP-181), which is elevated in AD, and is considered a marker of tangle formation [28]; and (c) total tau protein (τT) is a measure for N, which is increased in AD, and is a non-specific marker of neuronal and/or axonal degeneration [29]. The Aβ42/Aβ40 ratio may be preferred to Aβ42 alone since it appears to be a superior diagnostic tool compared with the latter [30]. Plasma Aβ42/Aβ40 levels are totally modified already during the pre-symptomatic phase; this explains why biomarkers like CSF Aβ42/Aβ40 detect Aβ pathology in cognitively normal subjects with comparable accuracies with cognitively abnormal people [31]. P-tau217 is dignified as the strongest among p-tau markers (p-tau181, p-tau 231, p-tau205). CSF p-tau217 is a stronger diagnostic tool than p-tau181 (area under the receiver operator characteristic curve (AUC), 0.943 vs. 0.914, *p* = 0.026) [32]. Simultaneously, CSF p-tau217 levels distinguish AD from other dementias, with higher accuracy than p181. Both plasma p-tau181 and p-tau217 precisely anticipate future MCI transformation to AD dementia (2 to 6 years) [33,34]. However, p-tau217 augments in the asymptomatic stage and alters with the progression of AD, permitting the prediction and early diagnosis of AD, while greater p-tau217 levels propose a rapid cognitive impairment [35]. Given the above privileges, p-tau217 is a proper biomarker concerning the T in the peripheral A-T-N-X framework. Importantly, plasma p-tau231 may be altering lightly before the other p-tau markers [36]. CSF p-tau217 showed the highest fold-change increases in symptomatic phages of the disease, while CSF p-tau231 untimely arrested the Aβ modifications in the preclinical stage. A key outcome of this study is that CSF p-tau231 is already significantly elevated before definite Aβ pathology. CSF p-tau231 was significantly associated with Aβ PET confinement in brain areas that are commonly impaired early in the AD, such as the medial orbitofrontal, precuneus, and posterior cingulate cortices in cognitively unimpaired subjects [37]. With a sensitivity and specificity at the level of ≥90%, they are useful in identifying the “AD neurochemical fingerprint” in atypical [38,39,40] or mixed cases [41,42], as confirmed with PET imaging.

The CSF contains more than 40 different endogenous APP and Aβ peptides, including alterations, that have been found thus far [40]. As a result, these endeavors yield more precise measurements of Aβ peptides in blood or CSF, but they may also identify distinct Aβ species, which prove advantageous in screening potential biomarkers for AD. Using mass spectrometry and the strong selectivity of anti-Aβ antibodies, for instance, Vigo-Pelfrey et al. [41] were able to determine the molecular mass with great accuracy, indicating the multiplex nature of Aβ peptides in the CSF and publishing several N- and C-terminal variants of Aβ. Additionally, in AD dementia and prodromal AD patients, the IP-MS approach measured elevated levels of CSF synaptosomal-associated protein 25 (SNAP-25) and synaptotagmin-1 (SYT1) [42]. Interestingly, reduced levels of SYT1 and SNAP-25 in cortical regions of the AD brain suggested that a group of synaptic proteins that include different regions of the synaptic unit would be useful in clinical research on the significance of synaptic degeneration and dysfunction in AD pathogenesis. These results demonstrated the efficacious method for detecting low abundance proteins, primarily from the central nervous system, or different Aβ peptides as an AD biomarker.

#### Overview of Fluid Biomarkers in Clinical Trials

The A,T,N Research Framework incorporates biomarkers into the diagnosis process of AD and has applications in clinical trials and medication development. The FDA’s staging approach for AD makes it easier to develop drugs for the predementia phases of the disease and incorporates biomarkers into it [43], as Table 1 shows. Diagnostic biomarkers provide precise diagnoses and enable the classification of a disease based on the existence or lack of a certain pathophysiological state. In order to maximize the establishment of a drug–placebo difference, predictive biomarkers can be utilized to enrich populations and indicate the development of the disease. Treatment response prediction is made easier with the use of predictive biomarkers. Pharmacodynamic or activity biomarkers indicate the occurrence of a biological reaction in the patient receiving the therapeutic intervention. Safety biomarkers, such as biochemical, MRI, and electrocardiogram (ECG), are biomarkers for identifying unfavorable and unintended medication responses.

Donanemab is an immunoglobulin G1 monoclonal antibody that targets the insoluble, shortened form of β-amyloid that is only found in brain amyloid plaques and has been changed. Donanemab binds to the β-amyloid’s N-terminally shortened version, facilitating the phagocytosis of microglia that removes plaque. Following donanemab, there was a substantial correlation found between the Centiloid percent change in amyloid and changes in plasma pTau217 and glial fibrillary acidic protein. Furthermore, there was a strong correlation between the plasma levels of pTau217 and glial fibrillary acidic protein both before and after treatment. The investigation of donanemab (target class: amyloid-β) in the TRAILBLAZER-ALZ research was unable to find significant alterations in plasma Aβ42/40 ratio levels [43]. Due to its high diagnostic accuracy [44], robust correlations with tau and amyloid pathology [45], and equivalency with validated CSF biomarkers in head-to-head investigations [46,47,48], p-tau217 has emerged as one of the most promising blood-based biomarkers for AD. PET revealed that individuals receiving donanemab had altered brain amyloid plaques, which were correlated with lower plasma pTau217 and GFAP. Comparing donanemab to a placebo, the pace of cognitive impairment was slowed down. After 12 and 18 months in the therapy group, lecanemab (target class: amyloid-β) exhibits a substantial rise in CSF Aβ42 [49]. The Aβ40 concentrations did not change between the treatment and placebo groups. Moreover, at the 12- and 18-month follow-up, there was a decrease in the levels of CSF tTau, pTau181, and NRGN. Between the two groups, there was no documented difference in CSF NFL. Comparing lecanemab to a placebo, the plasma exhibited a greater Aβ42/40 ratio, decreased pTau181 and GFAP, and improved cognitive ratings. There was no discernible improvement in cognition for the gantenerumab group as compared to crenezumab (target class: amyloid-β). Patients on gantenerumab experienced a decrease in CSF pTau181, tTau, and NRGN. tTau, pTau181, Aβ40, CSF Aβ42, and Aβ40 were the main AD biomarkers that were not impacted by crenezumab [50]. Monoclonal antibodies with no discernible therapeutic benefit are being studied in phase 2 studies for semorinemab, gosuranemab, and tilavonemab (target class: Tau). Semorinemab demonstrated a decrease in CSF pTau181, pTau217, and tTau as well as a dose-dependent increase in plasma mid-domain tau, which is identified as their target engagement marker [51,52]. Additionally, gosuranemab demonstrated target engagement by reducing CSF N-terminal tau. On Tau PET, however, there was no change. After 12 weeks, tilavonemab raised plasma tTau, indicating target engagement, and decreased CSF-free tau in a dose-dependent manner [53]. The antisense oligonucleotide MAPTrx affects CSF tTau levels in a dose-dependent manner and targets the Tau target class. The antisense oligonucleotide MAPTrx (target class: Tau) has been demonstrated to affect CSF tTau concentrations in a dose-dependent manner [54]. Neflamapimod (target class: inflammation), a p38α kinase inhibitor, demonstrated a favorable trend for NRGN and decreased CSF levels of pTau181 and tTau in comparison to a placebo. Without improving episodic memory (HVLT-R), no statistically significant results were observed for NFL, Aβ42, or Aβ40 levels (Table 2).

As strong candidates for targeted, possibly individualized treatment, there are three potential molecular biomarkers: monoacylglycerol lipase (Mgll), apolipoprotein E4 (APOE4), and the phosphatidylinositol 3-kinase (PIK3), protein kinase (AKT), and glycogen synthase kinase-3β (GSK-3β) signaling pathways [55]. Crucially, monoacylglycerol lipase (Mgll) gene expression was directly suppressed following the activation of the aPKC-CBP pathway with metformin administration. This was demonstrated utilizing a transgenic mouse model, CbpS436A, where the aPKC-CBP pathway is defective. All things considered, metformin was able to restart the damaged aPKC-CBP pathway to suppress Mgll expression, thereby saving the 3 × Tg mice’s hippocampus neuronal differentiation and spatial memory deficits. Mgll levels were aberrantly elevated in these mice throughout the aging process. In this sense, Mgll is the best possible candidate biomarker to identify potential patients who satisfy metformin’s requirements and are in the early stages of AD. A possible precision treatment approach for AD is anti-APOE4 immunotherapy, which uses antibodies to target and neutralize the APOE4 protein [56]. Another tactic to combat the pathogenic effects of APOE4 is the development of tiny compounds that disrupt its domain contacts. The discovery of these proteases and the creation of inhibitors against them may help prevent APOE4 toxicity given that neuronal proteases that break down APOE4 are known to release neurotoxic fragments. Moreover, since APOE4 lowers APOE2 receptor levels, increasing APOE2 receptor expression may be a therapeutic strategy to promote the “protective” effects of APOE2 rather than the “toxic” effects of APOE4 [56].

The insulin resistance-induced disruption of the PI3K/AKT signaling pathway raises GSK-3β activity and causes tau hyperphosphorylation, which puts people at risk for AD [55]. It appears that the pathophysiology of AD depends on this modulation of the PI3K/AKT/GSK-3β pathway [57]. As a result, tailored medication to lower GSK-3β activity has emerged as a viable treatment for AD [58]. It is now shown that lithium, a mood stabilizer for mental illnesses, inhibits GSK-3β activity both directly and indirectly. While intrahippocampal Aβ injection-treated rats and rats overexpressing GSK-3β and human amyloid precursor protein saw significant reductions in neuropathology and cognitive issues following lithium administration, other murine models of AD showed no improvement [58]. Therefore, patients with aberrant GSK-3β activity may be the only ones for whom lithium treatment is effective against AD-associated cognitive impairments and neuropathology [59]. Although there is currently no approved GSK-3β-specific neuro-radiotracer for use in humans, one substance has demonstrated significant advancements in primate brain research.

### 3.2. Genetic Biomarkers

Genetic biomarkers have also played a significant role in the detection of people at risk for AD. Variants in the apolipoprotein E (APOE) gene have been strongly correlated with a high risk of AD, with the APOE ε4 allele being the most well-established genetic risk factor for late-onset AD [64]. The APOE ε4 allele has been consistently related with an increased risk of AD, with individuals carrying one copy of the allele having a threefold increased risk and those with two copies having a twelvefold increased risk compared to individuals with the more common ε3 allele. The APOE ε4 allele has also been linked to an earlier age of onset and faster disease progression, making it a crucial genetic marker for AD risk assessment [65].

The general population is frequently (95%) affected by sporadic AD, which manifests as late-onset AD (LOAD) in people over 65. Age, female sex, traumatic brain injury, depression, environmental pollution, physical inactivity, social isolation, low academic level, metabolic syndrome, and genetic susceptibility—primarily mutations in the ε4 allele of apolipoprotein E (APOE, 19q13.32)—are the main risk factors of sporadic AD [66]. The heritability of the condition can reach 60–80%. The familial form of genetic AD is autosomal dominant, early onset (EOAD) in people under 65 (affecting 1 to 5% of cases) and typified by mutations in particular genes, including presenilin 1 (PSEN1, 14q24.2), which has been found to be altered in up to 70% of cases of familial AD; presenilin 2 (PSEN2, 1q42.13); and the amyloid precursor protein gene (APP, 21q21.3) [67]. In the same line as APOE, recent genome-wide association studies ((GWAS) reported over 30 genetic loci (CLU, PICALM, CR1, BIN1, EPHA1, MS4A, ABCA7, CD33, and CD2AP) associated with late-onset AD risk, highlighting the polygenic nature of the disease [68,69,70,71]. These loci include genes involved in various biological pathways, such as immune response [72], lipid metabolism, and synaptic function, providing new insights into the pathophysiology of AD. While these genetic variants individually confer only modest increases in risk, their cumulative effects can significantly impact an individual’s likelihood of developing the disease [68,71]. Rare variants (allele frequency) that influence the risk for LOAD have also been detected in several genes, including TREM2, PLD3, UNC5C, AKAP9, ADAM10, and ABI3. Genetic testing for these variants captured people at risk for AD and informed personalized prevention and treatment strategies.

The genetic basis for amyloid precursor protein profusion in Trisomy 21, also known as Down syndrome (DS), is EOAD. Because to the overabundance of Aβ and the amyloid precursor protein, by the mid-40s, all DS patients have enough ADNPC to meet the neuropathological criteria for an AD diagnosis [73]. The same level of genetic penetrance as in autosomal dominant AD (ADAD) is consistent with the age at onset and mortality in DS. With a mid-50s typical age of onset for clinical symptoms, the lifetime probability of dementia is 95% in DS [74]. Increased levels of peripheral proteins, including Aβ40; Aβ42; MMP-1, 3, and 9; proNFG; and inflammatory mediators like IFN-γ, TNF-α, IL-6, IL-10, and IL-1 were among the changes in plasma biomarkers found in DS [75]. There is, however, always some degree of doubt regarding the precise timing of these changes as well as whether the altered biomarkers are caused by inherited AD or DS. Notably, Aβ1–42/1–40 levels in cerebrospinal fluid decreased, hippocampi shortened, plaque burdens increased, cortical metabolism slowed, and plasma phospho-tau181 levels rose sooner in individuals with Down’s syndrome and ApoE4 [24,76]. There were no differences in CSF p-tau181, total tau, or both fluid NfL levels [76].

One key application of genetics for precision medicine in AD is the development of polygenic risk scores (PRS). Deep learning analyses of PRS combine information from multiple genetic variants associated with AD risk to generate a single numerical score that reflects an individual’s overall genetic susceptibility to the disease [77]. Several studies have shown that PRS can effectively stratify individuals into different risk categories, with higher scores correlating with an increased likelihood of developing AD. By identifying individuals at high genetic risk, PRS can facilitate targeted screening and preventive interventions to mitigate disease progression. For example, people with a positive family history of AD and the APOE4 gene variant could benefit from lifestyle modifications, a healthy diet, and regular exercise to decrease their risk of AD.

Furthermore, genetics can inform the development of personalized treatment strategies for AD patients based on their genetic profiles. Pharmacogenomic studies have identified genetic variants that influence individual responses to AD medications, such as cholinesterase inhibitors and memantine. Genes implicated in AD risk through GWAS and other studies can provide valuable insights into disease mechanisms and pathways that may be targeted for therapeutic interventions. For example, genes involved in amyloid beta metabolism, tau phosphorylation, and neuroinflammation have emerged as promising candidates for drug discovery efforts aimed at slowing or halting AD progression.

By genotyping patients for these variants, clinicians can tailor drug dosages and selection to optimize therapeutic outcomes and minimize adverse effects. Additionally, genetic testing can help identify individuals who may benefit from emerging precision therapies targeted at specific genetic subgroups, such as gene editing technologies or gene-based immunotherapies.

### 3.3. Neuroimaging Biomarkers

In addition to genetic testing, precision medicine for AD also involves the use of advanced imaging techniques, such as positron emission tomography (PET) scans and magnetic resonance imaging (MRI). These imaging technologies open the horizons for researchers to visualize changes in the brain associated with AD, such as the buildup of amyloid plaques and neurofibrillary tangles. Compared to structural MRI T1-weighted imaging, diffusion-weighted imaging (DWI) may offer extra and/or complementary information on the cortical thickness of presymptomatic subjects with familial AD [78]. Studies have suggested that DWI changes may be a better indicator of early progressive cognitive decline than macrostructural atrophy, whereas alterations in the white matter of the brain could be used as biomarkers for the conversion of MCI in AD [79]. Certain monoclonal antibodies cause amyloid-related imaging abnormalities (ARIA), which must be monitored with MRI throughout clinical trials to ensure the safety of these treatments.

By utilizing these imaging techniques, researchers can track disease progression, monitor treatment responses, and identify individuals who may benefit from early intervention. For example, amyloid PET scans can detect the existence of Aβ plaques in the brain, which are a hallmark neuropathological feature of AD [80]. In vivo PET studies with [18F]-labeled amyloid tracers detected moderate–frequent neuritic amyloid plaques with higher sensitivity (88–98%) and specificity (80–95%) compared to postmortem [81,82,83]. An accumulation of Aβ may be detectable by amyloid-specific imaging agents for positron emission tomography-computed tomography (PET/CT) as early as 15 years prior to the onset of AD symptoms, whereas the next most sensitive metric, cerebral hypometabolism (FDG-PET/CT), is identifiable only 10 years prior to symptom onset. Hypometabolism is thought to be a consequence of synaptic impairment during cell death. FDG PET measures metabolic activity, which is generally reflective of synaptic activity and neuron activation [84]. With time, FDG-PET was discovered to be a more accurate and focused biomarker for AD early diagnosis (sensitivity 95%, specificity 71% in people with mild AD) [22]. PET imaging uses ligands binding to microglial proteins to measure microglial activation; increased microglial activity has been observed in the medial temporal, occipital, and parietal lobes in AD dementia patients. Aβ PET/CT is thought to precede by 10 years the declines in even the most sensitive cognitive metrics, including episodic memory [85]. However, amyloid PET may lack specificity for distinguishing amyloid plaques and tau neurofibrillary tangles. Recently, [11C]-Pittsburgh compound B (PIB) amyloid PET reported high (89–100%) sensitivity and (88–98%) specificity in identifying intermediate–high AD neuropathologic change (ADNC) [86,87]. Individuals with high levels of amyloid may be candidates for clinical trials testing new treatments aimed at reducing amyloid buildup. Table 3 summarized the utility in research contexts, clinical practice, and trials of neuroimaging AD biomarkers.

### 3.4. Proteomics

The process of developing novel biomarkers typically involves three distinct phases: the identification phase, often known as screening; the validation phase; and the verification phase. One innovative area of AD biomarker research that is rapidly expanding in the field of AD precision medicine is mass spectrometry (MS)-based proteomic technology [88]. In the past ten years, the field of MS-based quantification in proteomics has been dominated by gel-free approaches (such as stable isotope labeling or employing label-free methods) in addition to gel-based techniques (such as 2D-PAGE and 2D-DIGE) [88]. Numerous candidate proteins that may serve as MCI or AD biomarkers have been identified using iTRAQ in conjunction with tandem mass spectrometry and multidimensional liquid chromatography [89,90]. These proteins were discovered to be involved in numerous biological pathways and processes, including oxidative stress response, inflammatory and immunological response, and Aβ metabolism. In addition, new technologies like SWATH-MS will be used to increase the likelihood of AD biomarkers even more. A particular further variation of data-independent acquisition (DIA) techniques, SWATH-MS is gaining popularity as a technology that combines quantitative consistency and accuracy with deep proteome coverage capabilities [91]. In addition to quantitative proteomics, the creation of assays for measuring specific post-translational modifications of proteins, like two-dimensional gel electrophoresis (also known as Western blotting or 2D-Oxyblot), has revealed the presence of specifically carbonylated proteins in the serum and hippocampi of triple transgenic mice modeling Alzheimer’s disease (3 × Tg-AD) at an early age [92,93]. According to this study, oxidative stress may be a key factor in the development of AD, and the oxidized proteins found in the serums could serve as early-stage AD biomarkers. Similar findings were made with MCI sufferers’ elevated serum protein carbonylation levels [94]. As a supplementary means of obtaining such extensive data, the proteomic technique is relatively new and more sophisticated for a protein biomarker analysis.

A tailored mass spectrometry method for protein quantification, such as multiple reaction monitoring (MRM) or selected reaction monitoring (SRM), is emerging as a means of bridging the gap between biomarker discovery and clinical validation. Assays for highly multiplexed molecular replacement modeling (MRM) can be easily set up to verify many candidates at once, making it easier to create biomarker panels that have the potential to improve specificity [95]. MRM’s capacity to quickly and continuously monitor only for the particular ions of interest can improve the lower detection limit for peptides. Stable isotopes combined with an MRM analysis provide multiplexing capacity and improve quantification reliability [95]. Given that AD is a complex illness, a panel of proteins is a better choice for an AD biomarker. As a result, MRM is a useful method for confirming potential biomarker candidates for AD and other potential real-world uses. MRM has been used in a number of investigations to find CSF-based protein biomarkers of AD [96]. The parallel reaction monitoring (PRM) approach has also been utilized in addition to MRM to assess potential biomarker candidates for AD [97]. Similar to the SRM technique, PRM offers the advantage of obtaining entire fragment spectra as opposed to a selection of preselected fragments; quantitation and high sensitivity are preserved, while interfering signals are prevented [98]. This would allow for the monitoring of other biochemical processes and proteins, including those in the innate immune system, secretory vesicles, and synapses, which are not directly linked to the accumulation of Aβ.

Owing to the low abundance and broad dynamic range of Aβ peptides, sample preparation is necessary prior to an MS analysis in the most frequent experimental approach used to quantify Aβ peptides in blood or CSF. There are numerous techniques currently available to concentrate and purify the Aβ peptides, including immunodepletion, size exclusion, ultrafiltration, immunoprecipitation, solid-phase extraction, and liquid–liquid extraction [98]. A recent paper employing IP in conjunction with the SRM-MS approach found that the concentration of plasma Aβ42 corresponded with the CSF Aβ42/Aβ40 ratio and was a strong predictor of the sensitivity and specificity of high brain Aβ. In a similar vein, the amyloid-β precursor protein (APP) 669–711/Aβ42 and Aβ40/42 ratios, together with their composites, have been shown to predict Aβ brain load at the individual level with 90% accuracy for an AD diagnosis, as established with PET [99]. Notably, amyloid-degrading enzymes most likely regulate Aβ in normal APP and Aβ metabolism [97]. Depending on the distinct APP breakdown mechanisms, different lengths of Aβ peptides can be found in vivo [100]. Interestingly, these methods not only provide a more precise measurement of Aβ peptides in blood or CSF, but they can also identify different species of Aβ, which is useful in screening potential biomarkers for AD. To illustrate the complex nature of Aβ peptides in the CSF, Vigo-Pelfrey et al. used mass spectrometry in conjunction with the high selectivity of anti-Aβ antibodies to measure the molecular mass with great accuracy [41]. They also identified multiple distinct N- and C-terminal variants of Aβ. Using the IP-MS approach, which has also been used to quantify the protein levels in the CSF, it was discovered that AD dementia and prodromal AD cases had significantly higher CSF levels of both synaptosomal-associated protein 25 (SNAP-25) and synaptotagmin-1 (SYT1) [101]. Importantly, cortical areas in the AD brain have lower levels of both SNAP-25 and SYT1 [102]. This suggests that a set of synaptic proteins covering various synaptic unit components may be useful tools in clinical studies on the significance of synaptic dysfunction and degeneration in AD pathogenesis. This approach has the benefit of identifying low abundance proteins, particularly from the central nervous system, or different Aβ peptides as a target biomarker of AD that may be used for precise AD diagnosis and treatment.

### 3.5. Metabolomics

The most recent omic platform, metabolomics, has enormous promise for the identification and treatment of neurodegenerative illnesses. This is a result of environmental factors as well as changes in transcription, genetics, and protein profiles. Two analytical platforms that are frequently employed for detection are mass spectrometry (MS) and nuclear magnetic resonance (NMR) spectroscopy. For metabolite structural testing, NMR is an especially useful technique. When it comes to identifying and quantifying intricate biological systems, an MS-based method is sensitive [65]. The field of metabolomics comprises many methodologies, such as fluxomics, lipidomics, untargeted metabolomics, and targeted metabolomics [89]. Hundreds of metabolites are measured by untargeted metabolomics to find metabolic fingerprints associated with a specific disease state or phenotype. For research projects, when the impacted metabolic pathways are unknown, this method—which gives relative changes in metabolites—is helpful. Quantitative measurements of a specific group of metabolites in an interesting pathway, such as glycolysis or the TCA cycle, are provided by targeted metabolomics. Lipidomics quantifies alterations in lipid profiles and necessitates specialized procedures for metabolite identification and analysis that are insoluble in water. Fluxomics, which is conducted in cells or in vivo, uses stable isotope tracers to give a dynamic, as opposed to static, assessment of metabolic processes. More study is vital; however, metabolomic investigations using biological samples from people with AD and MCI revealed metabolic alterations in plasma, CSF, and saliva that are linked to preclinical and clinical AD.

### 3.6. Epigenomics

Any process via which the environment can modify a phenotype without changing the genotype is known as epigenetic alterations, and they may necessitate a signaling cascade from the production of transcription factors. There are currently over twenty recognized epigenetic mechanisms, such as DNA methylations, genomic imprinting, noncoding RNAs (ncRNAs), post-translational modifications of histones (PTM-Hs) that alter gene expression by activating or repressing it, and a variety of confounding variables associated with changes in the environment.

The dysregulation of miRNA, small ncRNAs of 20–22 nucleotides in length, which regulate the half gene expression post-transcriptionally by binding to the 3′ untranslated region (UTR) of target mRNAs, is implicated in various neurodegenerative disorders, including AD, where miRNAs can modulate the expression of genes involved in amyloid-beta metabolism, tau phosphorylation, neuroinflammation, and synaptic dysfunction [103]. Through the sequential activity of cleavage enzymes BACE1 and γ-secretase, miRNAs can modify the processing of amyloidogenic APP into neurotoxic Aβ-42/40 and p-tau aggregates by modulating the target genes. Tauopathy and the development of amyloid plaques are encouraged when the CAMK4 gene, which controls synaptic activities in neuronal cells, is inhibited by microRNAs. Likewise, the disruption of the Dicer/Drosha complex results in the termination of miRNA production and is linked, albeit indirectly, to the deregulation of DNMT enzymes and, consequently, DNA methylation. In AD brains, the ADAM10 gene is implicated in APP processing and Aβ-amyloidosis; it is overexpressed due to particular miRNA molecules inhibiting the gene. By integrating omic data with bioinformatics analyses, researchers can identify potential target genes, regulatory networks, and signaling pathways modulated by dysregulated miRNAs in AD. An exhaustive review of 26 studies demonstrated the potential of circulating miRNAs (miR-107, miR-125b, miR-146a, miR-181c, miR-29b, and miR-342) as blood biomarkers for differentiating AD from controls [102]. Among 8098 quantified miRNAs, only 23 were significantly expressed in two or more studies. MiR-29a/b, miR-34a, and miR-125b have been implicated in amyloid-beta metabolism and tau phosphorylation, contributing to the accumulation of toxic protein aggregates and neuronal dysfunction in AD. miR-132, miR-146a, and miR-124 have been shown to modulate neuroinflammation and immune responses in AD by targeting pro-inflammatory cytokines and signaling pathways. Interestingly, miR-107 has been found to be associated with the dysregulation of proteins involved in aspects of AD pathology as well as being consistently down regulated in AD brains [102]. Therefore, the differential expression of these and other miRNAs in AD brains and biofluids underscores their potential as biomarkers for disease diagnosis, prognosis, monitoring disease progression, and developing personalized treatment strategies for AD.

### 3.7. Exosomes

Exosomes are important for cellular communication, the removal of harmful proteins from cells, and the spread of cellular pathogens to neighboring cells. They are composed of proteins, messenger RNAs (mRNAs), and microRNAs (miRNAs) that are indicative of their cellular origin. NEBs, or neuron-derived exosomes, circulate in the interstitial space in both the brain and the periphery and are found in bodily fluids such as blood, CSF, and urine [91]. In AD instances, it might act as a sign of underlying CNS abnormalities. Targeted analyses of endothelial, astrocyte, or neuronal cells can be carried out with the appropriate antibodies [92]. Numerous proteins found in neural-derived plasma exosomes have been linked to preclinical AD [93], and cargo proteins from exosomes formed from plasma astrocytes in AD have also been found [94]. Remarkably, when compared to stable MCI cases and normal control participants, changes in plasma NED levels of p-ptau, Aβ42, neurogranin, and repressor element 1-silencing transcription factor were observed among AD and MCI cases that transitioned to AD within 36 months [95]. Furthermore, it appears that miRNAs released from exosomes are linked to a number of neurodegenerative disorders, including AD, which is characterized by the buildup of Aβ plaques and hyperphosphorylated tau proteins [96]. The possible use of miRNAs as diagnostic biomarkers has been spurred by particular patterns of exosomal miRNAs from human bodily fluids, such as plasma and CSF [97,98]. The hunt for exosome-based biomarkers for AD and other neurodegenerative illnesses is further promoted by these outcomes.

## 4. Discussion

One of the key challenges in the field of biomarkers for precision medicine in AD is the heterogeneity of the disease and the variability of biomarker levels across individuals. AD is not a single entity but rather a complex syndrome with multiple underlying pathologies, including amyloid and tau pathology, neuroinflammation, oxidative stress, and synaptic dysfunction. As a result, a combination of biomarkers that capture the different aspects of the disease pathology may be needed to provide a comprehensive assessment of an individual’s disease status and guide personalized treatment decisions.

Moreover, the availability and accessibility of biomarker testing for AD remain limited, with many biomarker assays being expensive, invasive, or not widely available in clinical settings. For example, biomarker assays targeting tau phosphorylation on Thr217 could be different because of their composition (e.g., the use of antibodies targeting multiple or single phosphorylation sites), which may result in detached correlations with pathology. Therefore, it is a priority to confirm their associations with the core biomarkers of AD and their comparative diagnostic performance. It is also of great importance whether various plasma p-tau217 biomarkers are in line when capturing AD pathology in vivo, which will increase confidence in their future clinical use. As the field of biomarker research continues to advance, efforts are underway to develop standardized protocols for biomarker testing, establish reference ranges for biomarker levels, and validate biomarker assays for clinical use. Collaborative research initiatives, such as the Alzheimer’s Disease Neuroimaging Initiative (ADNI) and the European Medical Information Framework for Alzheimer’s Disease (EMIF-AD), are working to accelerate the development of standardized protocols for the collection, analysis, and interpretation of precision medicine AD biomarker data to ensure consistency and reliability across different research studies and clinical settings.

Despite these challenges, biomarkers hold great promise for revolutionizing the diagnosis, treatment, and management of AD. By enabling the early detection of the disease, tracking its progression, and assessing treatment response, biomarkers can empower clinicians to deliver personalized care tailored to the individual needs of each patient. In the era of precision medicine, biomarkers will play a crucial role in guiding therapeutic decisions, optimizing treatment outcomes, and ultimately improving the quality of life for individuals affected by AD.

While genetics for precision medicine in AD holds great promise, several challenges remain to be addressed in order to fully realize its potential. One major challenge is the interpretation of genetic data and the translation of research findings into clinically actionable insights. Genomic data are complex and multifaceted, requiring sophisticated analytical tools and expertise to extract meaningful information about disease risk and treatment responses. Improvements in data integration, bioinformatics, and artificial intelligence technologies will be essential for accelerating the translation of genetic AD research into clinical practice.

One of the key weaknesses in metabolomic research is the heterogeneity of AD patients. AD is a complex and heterogeneous disease with multiple subtypes and clinical manifestations. Metabolomic studies have shown that AD patients exhibit distinct metabolic profiles compared to healthy individuals, but there is also considerable variability within the AD population. This heterogeneity can complicate biomarker discovery efforts and limit the generalizability of findings across different cohorts.

Another limitation is the need for large-scale genetic studies with diverse and representative populations to ensure the generalizability of genetic findings across different ethnic groups and environmental contexts. The majority of the published AD biomarker data has been derived from highly educated, non-Hispanic white cohorts, and these biomarkers have not yet been extensively tested in broadly representative populations. Relationships among biomarkers, genetic variants like APOE ε4, and clinical outcomes may differ by race/ethnicity. Most genetic studies in AD to date have been conducted in populations of European ancestry, leading to a limited understanding of genetic risk factors in non-European populations. Definitive observational studies with more representative cohorts are needed to assess natural history relationships among biomarkers, genetics, comorbidities, and clinical outcomes. Furthermore, randomization rates and eligibility rates for AD clinical trials vary disproportionately by race/ethnicity, education, and socioeconomic status. Many metabolomic studies in AD have been small-scale and exploratory in nature, leading to inconsistent findings and conflicting results. The replication of findings in independent cohorts is essential for establishing the robustness and validity of potential biomarkers. Large-scale, multi-center studies with standardized protocols and rigorous validation procedures are needed to increase the reliability and reproducibility of metabolomic data in AD research. Efforts to increase diversity in genetic research through collaborative initiatives and data sharing will be critical for advancing personalized medicine approaches that are inclusive and equitable.

In addition, ethical and privacy considerations must be carefully addressed in the implementation of genetics for precision medicine in AD. Genetic testing raises concerns about data security, confidentiality, and informed consent, particularly in the context of sensitive information related to neurodegenerative diseases. Robust regulatory frameworks and guidelines are needed to safeguard patient rights and ensure the responsible use of genetic data in clinical practice.

One of the interesting findings of using amyloid PET imaging as a precision biomarker for AD is the interpretation of amyloid PET results. While amyloid PET imaging has high sensitivity and specificity for detecting amyloid plaques, the presence of amyloid deposition does not always correlate with the presence of AD pathology or cognitive impairment. This has led to the concept of “amyloid positivity” and “amyloid negativity” in the context of AD diagnosis, with some individuals showing amyloid deposition but no cognitive impairment and vice versa. To address this issue, researchers have been exploring the use of multimodal imaging approaches that combine amyloid PET imaging with other biomarkers, such as tau PET imaging and structural MRI, to improve the accuracy of AD diagnosis and prognosis. For example, a study by Jack et al. found that combining amyloid PET imaging with tau PET imaging and structural MRI improved the prediction of cognitive decline in individuals with mild cognitive impairment. This multimodal approach holds promise for improving the precision of AD diagnosis and prognosis and for guiding personalized treatment strategies. Another challenge to using amyloid PET imaging as a precision biomarker for AD is the cost and availability of this imaging technique. Amyloid PET imaging is currently limited to specialized imaging centers and may not be accessible to all individuals with suspected AD. However, recent advances in PET technology and radiotracer development have led to the commercialization of amyloid PET tracers, such as florbetapir (Amyvid) and flutemetamol (Vizamyl), which are approved by the FDA for clinical use. In addition, efforts are underway to develop more affordable and widely available amyloid PET tracers, which could expand the use of amyloid PET imaging as a precision biomarker for AD.

## 5. Research Gaps

There is a lack of certified biofluid reference methods and materials (except for cerebrospinal fluid [CSF] amyloid beta [Aβ]42, where these are available).The RNA and exosome isolation and downstream miRNA detection, quantification, and normalization methods varied between studies, such as enzyme-linked immunosorbent assays (ELISA), Western blotting, and mass spectrometry (S, showing conflicting results).No comprehensive biofluid analyses exist for CSF and blood levels of multiple inflammatory markers, along with Core 1 and 2 biomarkers.In order to empower cohorts for maximized therapeutic effects in clinical trials, understanding the predictive and prognostic value of omic signatures relevant to clinical trajectories is crucial.Despite the efforts, PET, CSF, and blood biomarkers remain less sensitive compared with neuropathologic examination for the detection of early/mild AD neuropathologic change (ADNPC). Disease staging by PET (or fluid biomarkers) is not equivalent to neuropathological staging; for example, tau PET ligand uptake in different Braak areas is not equivalent to Braak neuropathological staging. While the sensitivity limits of biomarkers could be appraised as a disadvantage, they could also be appraised as a strength because abnormal Core 1 biomarkers indicate that ADNPC more generally than just neuritic plaques alone is very likely present.Thoroughly studied biomarkers are not available for all relevant diseases; there is a high uncertainty of other co-pathologies in addition to AD in any individual or what the proportional disease-specific burden is among various pathologic entities.The proportion of the cognitive deficit observed in a single patient that is attributable to AD versus other neuropathologic pathologies is difficult to quantify. Only probabilistic rates can be calculated based on combinations of biomarker results and clinical evaluation.

## 6. Future Steps

Future protocols for clinical trials should rigorously include more representative cohorts. True epidemiological and real-world data studies of biomarker properties in representative groups are crucial to determining relationships that are valid at the population level. A better understanding of the longitudinal intra-individual biological and disease-associated variability; the potential impact of clinical confounders and biological factors, including race and ethnicity, peripheral neuropathies and other neurologic diseases, BMI, and kidney disease; and the relative effects on the clinical performance of plasma Aβ42/Aβ40, p-tau, NfL, and GFAP in large cohorts is needed. In order to minimize referral bias, prospective studies in the general population would minimize the risk of overestimating the power of ApoE4.Longer clinical trials are needed to show the lowering rate of brain volume loss as a result of the amyloid plaque removal.An international consensus of standard biofluid assays, tau PET quantification methods, and cutpoints is warranted. As in other diseases, the exact thresholds for abnormality may evolve over time as additional data inform the prognostic value.Advanced knowledge of various post-translational modifications of tau may enhance fluid-based biological staging. The integration of genomic and epigenomic data to ascertain the influence of epigenetic mechanisms in the setting of complicated disease phenotypes may be made possible by artificial intelligence methods.With an improved understanding of the role of immune/inflammatory processes, microglia, and astrocyte biology in AD pathogenesis, we foresee a more notable role for biomarkers in biological characterization and prognosis, especially if brain-specific modifications can be revealed in blood.Keeping in mind that clinical trials target mechanisms other than anti-Aβ immunotherapy, the effects of these interventions on biomarkers and clinical outcomes should be included in future diagnostic AD criteria.By identifying miRNA targets, regulatory networks, and signaling pathways implicated in disease pathogenesis, researchers can develop small molecule inhibitors, antisense oligonucleotides, and gene therapies that modulate miRNA function, restore gene expression, and reverse neurodegeneration in AD.

## 7. Conclusions

Biomarkers of precision medicine for AD represent a transformative approach to healthcare that has the potential to revolutionize the diagnosis, treatment, and management of this devastating neurodegenerative disorder. By leveraging a combination of imaging, genetic, novel ultrasensitive immunoassays, mass spectrometry methods, metabolomics, and exosomes that show promise for fluid biomarkers, researchers are making significant strides towards personalized healthcare for individuals with AD. The integration of biomarkers into clinical practice holds the promise of improving diagnostic accuracy, prognostic assessment, and therapeutic decision-making for affected individuals, ultimately paving the way for more effective and individualized treatments for AD. As we continue to unravel the complex pathophysiology of AD and identify new biomarkers for precision medicine, we move closer towards a future where personalized care for individuals with Alzheimer’s disease becomes a reality.

## Figures and Tables

**Figure 1 jcm-13-04661-f001:**
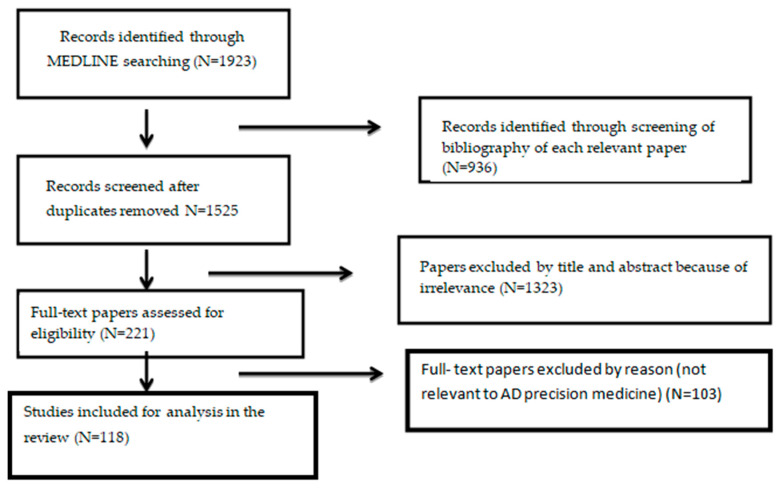
A flowchart of the study selection.

**Table 1 jcm-13-04661-t001:** Role of biomarkers for each phase of AD drug development based on FDA guidelines.

Biomarkers	Screening	Phase 1	Phase 2	Phase 3
Diagnostic biomarkers of AD Low CSF Aβ42 or CSF Aβ42/t-tau ratio or Aβ42/ptau ratio or positive amyloidPET	Demographic data based such as CAIDE dementia risk score, ADAS-Cog symptomatic AD A+T+ is mandatory and exclusion of comorbidities should be conducted			
Predictive biomarkers			Tau PET used to determine whether AD patients are more likely to benefit from anti-tau treatments	
Prognostic biomarkers:Sort people based on likelihood of illness or include more patients in trials			Tau PET to determine which AD patients are most likely to experience cognitive decline more quickly ApoE-4 carriers in immunotherapy studies as a prognostic marker for ARIA	Tau PET to determine which AD patients are most likely to experience cognitive decline more quickly ApoE-4 carriers in immunotherapy studies as a prognostic marker for ARIA
Pharmacodynamic biomarkers: (i) Target engagement (ii) Disease modification (atrophy on MRI, hypometabolism on FDG PET, or increases in total tau in the CSF)			Phase 2′s essential result for moving on to Phase 3	Essential outcome for the intervention to be classified as a DMT
Safety biomarkers		In immunotherapy regimens, liver function and other laboratory tests, an ECG, and an MRI are used to check for ARIA.	In immunotherapy regimens, liver function and other laboratory tests, an ECG, and an MRI are used to check for ARIA.	Liver function and other laboratory tests, ECG, MRI to monitor for ARIA in immunotherapy programs

ApoE: apolipoprotein E, ARIA: amyloid-related imaging abnormalities, CSF cerebrospinal fluid, ΕCG: electrogardiogram, FDG fluorodeoxyglucose, MRI: magnetic resonance imaging, PET: positron-emission tomography.

**Table 2 jcm-13-04661-t002:** Overview AD biomarkers in clinical trials.

Study Ref	Drug	Study Characteristics (Phase, Duration, n, Age Range)	Tools (Clinical Scales, Neuroimaging)	Biomarker Changes	Clinical/Neuropsychological Outcomes	Potential Relevance Both from Clinical and Biological Perspective
Fang et al. [44]	Buntanetap (Amyloid-β)	Phase 2 4 w N = 75	CDR-SB and MMSE scores	CSF Aβ40: NS vs. placebo CSF Aβ42: NS vs. placebo CSF tTau: NS vs. placebo CSF pTau: NS vs. placebo CSF sAPPa: NS vs. placebo CSF sAPPb: NS compared to placebo CSF sTREM2: NS vs. placebo CSF GFAP: NS vs. placebo CSF YKL-40: NS compared to placebo CSF complement 3: NS vs. placebo CSF NFL: NS vs. placebo CSF NRGN: NS vs. placebo ptau: NA Study not powered to measure statistically significant differences, trends were visible.	ADAS-Cog11: Better score vs. baseline WAIS: Better score vs. baseline MMSE: NS vs. baseline CDR-SB: NS vs. baseline	Buntanetap as exploratory biomarker showing anti-inflammatory function and synaptic integrity
Ostrowitzki et al. [45]	Crenezumab (Amyloid-β)	Phase 3 100 w N = 805 50–85	Amyloid PET or CSF	Discontinued due to earlier study not meeting primary endpoint	Discontinued due to earlier study not meeting primary endpoint	Crenezumab did not reduce clinical decline in early AD
Sims et al. [46]	Donanemab (Amyloid-β)	Phase 3 76 w N = 1800 60–85	Gradual and progressive change in memory; Tau PET and amyloid PET	Plasma pTau217: decreased (Log_10_ −0.2) vs. placebo	iADRS: Better score compared to placebo	Donanemab significantly slowed clinical progression at 76 weeks in those with low/medium tau and in the combined low/medium and high tau pathology group according to PET biomarkers
Mintun et al., Pontecorvo et al. [47,48]	Donanemab (Amyloid-β)	Phase 2 72 w N = 266 60–85	Gradual and progressive change in memory; positive Amyloid and Tau PET	Decreased Plasma pTau217 (Log_10_ −0.14) and GFAP: vs. placebo Plasma Aβ42/40, NFL: NS vs. to placebo	iADRS: Better score vs. to placebo ADAS-Cog13: Inconclusive CDR-SB/ADCS-iADL/MMSE: NS vs. placebo	Plasma biomarkers pTau217 and glial fibrillary acidic protein than placebo following donanemab might provide additional evidence of early symptomatic AD pathology change through anti-amyloid therapy.
Bateman et al. [49]	Gantenerumab (Amyloid-β)	Phase 3 116 w N = 1016 50–90	CSF tau/Aβ42 or amyloid PET scan	Decreased CSF tTau, pTau181, Aβ40: vs. to placebo Increased CSF Aβ42: vs. placebo Decreased CSF NRGN and NFL vs. placebo Plasma pTau181: decreased vs. to placebo Plasma Aβ42: Increased vs. to placebo CSF pTau181: −23.8% Plasma pTau181: −24%	CDR-SB: NS compared to placebo ADAS-Cog13: NS compared to placebo ADCS-ADL: NS compared to placebo	Gantenerumab led to a lower amyloid plaque burden than placebo at 116 weeks without clinical improvement.
Bateman et al. [49]	Gantenerumab (Amyloid-β)	Phase 3 116 w N = 982 50–90	CSF tau/Aβ42, amyloid PET scan	Decreased CSF tTau, pTau181, Aβ40 vs. placebo CSF Aβ42: increased compared to placebo CSF NRGN: decreased vs. placebo CSF NFL: decreased vs. placebo Plasma pTau181: decreased vs. placebo Increased plasma Aβ42 vs. placebo CSF pTau181: −23.8% Plasma pTau181: −21%	CDR-SB: NS compared to placebo ADAS-Cog13: NS compared to placebo ADCS-ADL: NS compared to placebo	Gantenerumab led to a lower amyloid plaque burden than placebo at 116 weeks without clinical improvement.
Van Dyck et al. [50]	Lecanemab (Amyloid-β)	Phase 3 78 w N = 1766 50–90	Positive biomarker amyloid	Increased CSF Aβ42: vs. placebo Decreased CSF tTau and pTau181 vs. placebo Decreased CSF NRGN vs. placebo CSF Aβ40: NS vs. placebo CSF NFL: NS vs. placebo Increased Plasma Aβ42/40 vs. placebo Decreased Plasma pTau181, NFL, GFAP vs. placebo CSF pTau181: ~30 pg/mL compared to placebo −16 pg/mL compared to baseline Plasma pTau181: ~0.8 pg/mL	CDR-SB: Better score vs. placebo ADAS-Co14: Better score vs. placebo ADCOMS: Better score vs. placebo ADCS_MCI-ADL: Better score vs. placebo	Lecanemab reduced markers of amyloid in early AD and lower cognitive decline
Lerner et al. [51]	Efavirenz (ApoE, Lipids and Lipoprotein Receptors)	Phase 1 52 w N = 5 55–85	MMSE CDR	Increased Plasma 24-OHC vs. baseline CSF Aβ40: NS compared to baseline CSF Aβ42: NS compared to baseline CSF tTau: NS compared to baseline CSF pTau181: NS compared to baseline	MoCA: NS compared to baseline	CYP46A1 activation by low-dose efavirenz increased brain cholesterol metabolism (as measured by high HC levels) in early AD
Wilkins et al. [52]	S-equol (growth factors and hormones)	Phase 2 4 w N = 40 50–90	COX/CS	Increased COX/CS compared to baseline	MoCA: NS compared to baseline	S-equol May acts as a direct mitochondrial target engagement biomarker
Vissers et al. [53]	DNL747 (antiInflammatory)	Phase 1 12 w N = 16 55–85	CSF Ab42 Amyloid PET	Decreased Plasma PBMC pRIPK1 vs. placebo	No clinical endpoints included	RIPK1 in the CNS as a potential therapeutic tool for AD
Prins et al. [54]	Neflamapimod (antiInflammatory)	Phase 2 24 w N = 161 55–85	CDR, MMSE; CSF Ab1–42, p-Tau, CT, MRI compatible with AD	Decreased CSF tTau, pTau181 vs. placebo CSF NRGN: NS compared to placebo CSF NFL: NS compared to placebo CSF Aβ40: NS compared to placebo CSF Aβ42: NS compared to placebo CSF pTau181: −2.1 pg/mL	HVLT-R/WMS immediate and delayed recall/CDR-SB/MMSE: NS compared to placebo	Neflamapimod treatment lowered CSF biomarkers of synaptic dysfunction but not improve the cognitive scores
Sullivan et al. [55]	3TC (lamivudine)	Phase 2 24 w N = 12 50–80	CSF GFAP CSF Aβ42/40 CSF pTau181 Plasma Aβ42/40 CSF NFL Plasma GFAP Plasma pTau181	CSF GFAP: decreased vs. baseline Plasma Aβ42/40: increased vs. baseline CSF NFL: NS compared to baseline CSF Aβ42/40: NS compared to baseline CSF pTau181: NS compared to baseline Plasma NFL: NS compared to baseline Plasma GFAP: NS compared to baseline Plasma pTau181: NS compared to baseline	MMSE: NS compared to baseline PACC-5: NS compared to baseline Attention, memory, naming, and EF tasks: NS compared to baseline	Decreased levels of AD and inflammatory biomarkers suggested positive effect of 3TC against MCI due AD
LaBarbera et al. [56]	CT1812 (Synaptic plasticity/neuroprotection)	Phase 1 1 w N = 3 50–80	MRI and Abeta PET scan	CSF Aβ oligomers: Increased compared to baseline	No clinical endpoints were included	The degree of Aβ oligomers alteration aligned with the exposure level of CT1812 supports the use of Aβ oligomers as a biomarker of target engagement
Van Dyck et al. [57]	(CT1812 Synaptic plasticity/neuroprotection)	Phase 2 30 w N = 23 50–85	Amyloid PET or Amyloid CSF	CSF Aβ40: NS compared to placebo CSF Aβ42: NS compared to placebo CSF tTau: NS compared to placebo CSF pTau: NS compared to placebo CSF NRGN: NS compared to placebo CSF synaptotagmin: NS vs. placebo CSF SNAP25: NS compared to placebo CSF NFL: NS compared to placebo	ADCS-ADL: High dose better scores compared to placebo ADAS-Cog11: NS compared to placebo MMSE: NS compared to placebo	No treatment effects relative to placebo from baseline at 24 weeks in neither SV2A nor FDG PET signal, the cognitive clinical rating scales, or in CSF biomarkers
Mummery et al. [58]	BIIB080 (MAPT_rx_) (tau)	Phase 2 61 w N = 46 50–74	CSF biomarkers	CSF tTau: decreased compared to placebo CSF pTau181: decreased compared to placebo CSF tTau/Aβ42: decreased compared to placebo CSF NFL: NS compared to baseline CSF NFH: NS compared to baseline CSF NRGN: NS compared to baseline CSF YKL-40: NS compared to baseline CSF pTau181: Ranging from 0 to ~−55% based on dose	RBANS Total score: NS compared to baseline MMSE Total score: NS compared to baseline NPI-Q/FAQ Total score: NS compared to baseline	MAPT_Rx_ reduce tau levels in mild AD
Shulman et al. [59]	Gosuranemab (Tau)	Phase 2 238 w N = 654 50–80	Positive for amyloid beta	CSF Unbound N-terminal tau: decreased in treatment compared to placebo CSF pTau181: Decreased in high dose treatment compared to placebo CSF tTau: Decreased in treatment compared to placebo CSF Aβ42: NS compared to placebo −7.1 pg/mL compared to baseline CSF pTau181: ~−25 pg/mL compared to placebo	CDR-SB/MMSE/ADCS-ADL/FAQ: NS compared to placebo group ADAS-Cog13: Significantly worse in treatment compared to placebo	No significant effects in cognitive and functional scores but reduced levels CSF Unbound N-terminal tau in gosuranemab group
Teng et al. [60]	Semorinemab (Tau)	Phase 2 73 w N = 457 50–80	Amyloid PET CSF tTau and pTau181	Plasma mid-domain tTau: increased compared to placebo CSF tTau: decreased from baseline CSF pTau181: decreased from baseline CSF pTau181 change: −9.7 pg/mL compared to placebo/−10.5 pg/mL compared to baseline	CDR-SB/ADAS-Cog13/RBANS/ ADCS-ADL/A-IADL-Q: NS compared to placebo	Semorinemab did not slow clinical AD progression
Monteiro et al. [61]	Semorinemab (Tau)	Phase 2 72 w N = 273 50–85	MMSE CSF Ab42 Amyloid PET	Increased plasmatTau, pTau217 vs. placebo Decreased CSF tTau, pTau217, pTau181 vs. placebo CSF N-term Tau: NS compared to placebo Plasma pTau217: ~+88 pg/mL CSF pTau217: ~−50% CSF pTau181: ~−12%	ADAS-Cog11: Better score compared to placebo ADCS-ADL/CDR-SB/MMSE: NS compared to placebo	No treatment effects on functional scales nor on amyloid biomarkers
Fleiser et al. [62]	Zagotenemab (Tau)	Phase 2 104 w N = 360 60–85	Progressive change in memory > 6 m Plasma pTau181, tTau, NFL	Increased plasma tTau, pTau181 vs. placebo Plasma NFL: NS compared to placebo Plasma pTau181: ~+15 pg/mL (low dose); ~+ 30 pg/mL (high dose)	iADRS/ADCS-iADL/ADAS-Cog13/CDR-SB/MMSE: NS compared to placebo	Zagotenemab did not slow clinical disease progression. Imaging biomarkers and plasma NfL without pharmacodynamic activity or disease progress.
Willis et al. [63]	Zagotenemab	Phase 1 64 w N = 24 54	tTau	Plasma tTau: NS compared to placebo	No clinical endpoints included	The pharmacokinetics of zagotenemab were typical for a monoclonal antibody. Meaningful pharmacodynamic differences were not observed.

AD: Alzheimer’s disease, ADAS-Cog: Alzheimer’s Disease Assessment Scale–Cognitive Subscale, ADCS-iADL: Alzheimer’s Disease Cooperative Study Activities of Daily Living Inventory instrumental subscale, CDR: clinical dementia rating, CSF: cerebrospinal liquid, MMSE: mini mental state examination, N = number, NA: non-available, NS: nonsignificant, RBANS: Repeatable Battery for the Assessment of Neuropsychological Status, vs.: versus, w: weeks.

**Table 3 jcm-13-04661-t003:** Utilityof neuroimaging AD biomarkers in research contexts, clinical practice, and trials.

Type of Neuroimaging Biomarker	Utilityt in Research Context	Utility in Clinical Practice and Trials
Structural MRI		Atrophy of the hippocampus or the surrounding medial temporal lobe regions
DWI		More indicative of early progressive cognitive change
Functional MRI	Less connection between the medial temporal regions and the posterior cingulate cortex.	Not recommended for routine clinical usage (high cost, limited spatial resolution)
FDG PET	Reflective of synaptic activity and neuronal activating	Deficits in regional cerebral blood flowpredicting conversion to AD in people with MCIElevated microglial activity as an inflammatory marker to monitor the anti-inflammatory effects of AD treatments
Amyloid PET		Recognizing the intermediate-high neuropathologic alteration of ADThe retention time of PiB indicates the change of MCI to AD.
Tau PET		Measures the fibrillar deposited form of the tau proteinto monitor in anti-tau trials

AD: Alzheimer’s disease, DWI: Diffusion-weighted imaging, FDG PET: 18F-fluorodeoxyglucose-PET, MRI: magnetic resonance imaging.

## Data Availability

No new data were created or analyzed in this study.

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
