# Peer review of "Biomarker-Based Precision Therapy for Alzheimer’s Disease: Multidimensional Evidence Leading a New Breakthrough in Personalized Medicine"

_jcm, 2024, doi:10.3390/jcm13164661_

Round 1

Reviewer 1 Report

Comments and Suggestions for Authors

First of all, I would like to thank the authors for their work. Improving diagnosis and monitoring strategy of Alzheimer's disease (AD) in its early symptomatic stages is necessary, more even since the approval of potentially disease modifying antiamyloid drugs. 

Understanding how to apply precision medicine both for diagnosis and monitoring of AD is clearly mandatory. 

Having said all this, I have some serious concerns about the manuscript itself: 

1) Introduction: is misleading. Diagnosis of AD is clinics-biological and this diagnosis is feasible and should be done in prodromal stages of AD, advances in guaranteeing universal access to core AD biomarkers are being done, independently if the syndrome is typical or atypical presentation. This is even more necessary at the current era of potentially disease modifying drugs. Not explanation of AD core biomarkers updated in the diagnosis criteria published just last week. 

CSF ATN biomarkers and PET are frequently used in clinical practice. Explain better promising results of plasma biomarkers, mainly ptau217 to be used as first screening measure with very good diagnosis accuracy. This seems the next huge step in the field, and that could help to become more feasible the early detection of patients that could benefit from FDA approved drugs. 

2) Material and methods: explain not only Mesh Terms, but as well the combination used  to research with them. Why you only used a database? Could you explain better the inclusion and exclusion criteria of the manuscripts selected (may be a flowchart could be helpful). Why did you select ptau181 when ptau217 is clearly more accurate to the diagnosis of MCI-AD or prodromal AD an ptau231 even more accurate for very early (presymtomatic stages of AD). Also please add info in the intro or discussion about the disturbing (from clinical perspective, but very interesting from research perspective) capacity to detect AD in preclinical stages. 

2) Figure 1: misleading. First step in clinical practice is NRL-NPS assessment for MCI diagnosis, later AD diagnosis must be done using core biomarkers and checking for comorbidities (no diagnosis of MCI-AD is done without biomarkers); soon (in some places they have already started) for first screening they are using plasma biomarkers (ptau217 mainly) and also for monitoring. EEG and fMRI don't seem to be at the same level of other biomarker for monitoring. Explain APOE status could influence inclusion or not in case of anti-amyloid treatments. 

3) AT(N) biomarkers are NOT neurodegenerative biomarkers. Only (N) is indicative of neurodegeneration, and is the less relevant of the three categories. Explain how AT(N) biomarkers are NOT only relevant for atypical but also typical clinical syndromes for its early diagnosis.

4) Overview fluid biomarkers: in the text or directly in the table 1  explain amount of changes in biomarkers and clinical/neurophsychological level for each drug and its potential relevance both from clinical and biological perspective. 

5) Genetic biomarkers: explain better how APOE status could influence the capacity of inclusion in anti-amyloid therapies, also explain the newly suggested APOE4 related semidominant concept (APOE E4 homozygous as semidominant AD, Fortea et al 2024). 

6) The most innovative part of the review related with -omic strategies, those that clearly could be more related with personalized medicine are the one that are less explained during the manuscript. They are really the sections that could add extra value, together with more critical lecture of the literature, explaining how this advances should be incorporated to clinical practice (in a summary table) and also research gaps in the field and future steps that must  be done in order to try to solve them. 

Author Response

REVIEWER  1

First of all, I would like to thank the authors for their work. Improving diagnosis and monitoring strategy of Alzheimer's disease (AD) in its early symptomatic stages is necessary, more even since the approval of potentially disease modifying antiamyloid drugs. 

Understanding how to apply precision medicine both for diagnosis and monitoring of AD is clearly mandatory. 

Having said all this, I have some serious concerns about the manuscript itself: 

1) Introduction: is misleading. Diagnosis of AD is clinics-biological and this diagnosis is feasible and should be done in prodromal stages of AD, advances in guaranteeing universal access to core AD biomarkers are being done, independently if the syndrome is typical or atypical presentation. This is even more necessary at the current era of potentially disease modifying drugs. Not explanation of AD core biomarkers updated in the diagnosis criteria published just last week

 OUR RESPONSE: Τhank you for the valuable comment. We clarified this issue, such as ‘’Recently the revised criteria proposed the biological diagnosis of AD based on CSF or plasma and imaging biomarkers that are subclassified according the proteinopathy or pathophysiological pathway (A, T1, T2, N, I, V,S)[9]. Core 1 biomarkers (α-amyloid Aβ42, phosphorylated tau pTau 181, p-tau217, p-tau231 and amyloid Pet) define the initial stage of AD that is detectable in vivo and can identify the presence of AD in both symptomatic and asymptomatic individuals[9]. Core 2 biomarkers (MTBR-tau243, p-tau205 non-phosphorylated mid-region tau fragments and Tau PET) become abnormal later in the evolution of AD and inform on the risk of short-term progression in people without symptoms.’’ (lines 53-69 page 2)

CSF ATN biomarkers and PET are frequently used in clinical practice. Explain better promising results of plasma biomarkers, mainly ptau217 to be used as first screening measure with very good diagnosis accuracy. This seems the next huge step in the field, and that could help to become more feasible the early detection of patients that could benefit from FDA approved drugs. 

 OUR RESPONSE: We explained  more the role  of plasma biomarkers :’’ Currently, the following biomarkers have sufficient accuracy to be diagnostic of AD: amyloid PET; CSF Aβ 42/40, CSF p-tau 181/Aβ 42, CSF t-tau/Aβ 42; or “accurate” plasma assays where “accurate” can be defined as accuracy that is equivalent to approved CSF assays in detecting abnormal amyloid PET in the intended-use population.’’(lines 72-76 page 2)

Regarding the ’’ P-tau217 is considered to be the most robust among p-tau markers (p-tau181, p-tau 231, p-tau205) . In the diagnosis of AD, the performance of CSF p-tau217 is better than that of p-tau181 (area under the receiver operator characteristic curve (AUC), 0.943 vs 0.914, P = 0.026)[27] . At the same time, CSF p-tau217 can distinguish AD from other neurodegenerative diseases with dementia, and the accuracy is superior to that p181. Both plasma p‐tau181 and p‐tau217 have been shown to accurately predict future cognitive decline in patients with MCI, and conversion to AD dementia in the subsequent 2 to 6 years [28,29]. However, p-tau217 rises in the asymptomatic phase and changes with the progression of AD, allowing prediction and early diagnosis of AD, while higher p-tau217 levels suggest a faster cognitive decline [30]. Regarding the above advantages, p-tau217 is an appropriate biomarker with respect to the T in the peripheral A-T-N-X framework.’’(lines 185-198 page 5)

2) Material and methods: explain not only Mesh Terms, but as well the combination used  to research with them. Why you only used a database? Could you explain better the inclusion and exclusion criteria of the manuscripts selected (may be a flowchart could be helpful). Why did you select ptau181 when ptau217 is clearly more accurate to the diagnosis of MCI-AD or prodromal AD an ptau231 even more accurate for very early (presymtomatic stages of AD). Also please add info in the intro or discussion about the disturbing (from clinical perspective, but very interesting from research perspective) capacity to detect AD in preclinical stages.

 OUR RESPONSE: ‘’Although the scope of this study did not extend to performing a systematic review, we applied the basic principles of a systematic review, but restricted to published peer-reviewed articles and a narrative analysis . A literature search was performed using Medical Subject Heading (MeSH) terms on the PubMed  including “Alzheimer’s disease”, “biomarkers”, “APOE”, “APP”, “GWAS”, “cerebrospinal fluid”, “polygenic risk score”, “Aβ42”, “τP-181”, ‘’ p-tau217 ‘’ , ‘’ptau231’’, ‘’ total tau protein‘’ and “precision medicine”. We combined search terms for relevance to the topic using Boolean operators. Only articles written in the English language were included. Although studies in the past 10 years were favored, there was no restriction regarding the year of publication. Through the snowballing process, I also screened the bibliography of each selected paper for potential additional studies to source the majority of the recent key evidence.Relevant studies, international guidelines, society recommendations, consensus reports, practice guidelines, and expert panel reports published through June 2024 were included.

2.1. Inclusion Criteria

The inclusion criteria were as follows: human, animal, studies published in English.

2.2. Exclusion Criteria

The exclusion criteria were as follows: (1) dementia syndromes beside AD, and (2) reviews, letters, editorials, abstracts, conference proceedings, and theses.3) Studies that did not present results were ruled out.’’ (lines 94-103 page 6, lines 104-117 page 7)

Regarding the ’’p-tau217 has emerged as one of the most promising blood-based biomarkers for AD based on good diagnosis accuracy [32], strong associations with amyloid and tau pathologies [33] and equivalence with established CSF biomarkers in head-to-head studies[34-36].’’ (231-235 page6)….. Both plasma p‐tau181 and p‐tau217 have been shown to accurately predict future cognitive decline in patients with MCI,  and conversion to AD dementia in the subsequent 2 to 6 years[25,26].

We also add info in the intro about the  capacity of biomarkers to detect AD in preclinical stages:’’Core 1 biomarkers such as α-amyloid Aβ42, phosphorylated tau (pTau 181), (p-tau217), (p-tau231) and amyloid PET define the initial stage of AD that is detectable in vivo and can identify the presence of AD in both symptomatic and asymptomatic individuals[9]. Considering their time of onset, plasma p-tau217and p-tau231 have been suggested as biomarkers of Aβ plaques , but this is not conceptually correct because of the coexistence of tau fragments.  Core 2 biomarkers (MTBR-tau243, p-tau205 non-phosphorylated mid-region tau fragments and Tau PET) become abnormal later in the evolution of AD and inform on the risk of short-term progression in people without symptoms. p-tau217, p-tau181, and p-tau 231 have been shown to rise originally at the start of Aβ accumulation before the change in tau-PET, and p-tau205 and t-tau begin to increase close to the onset of clinical symptoms [10].’’

Regarding the ptau231:’’ Importantly, p‐tau231 might be changing slightly earlier than the other p‐tau markers[27]. CSF p-tau217 had the highest fold-change increases in symptomatic stages of the disease, while CSF p-tau231 better captured the earliest Aβ changes in the preclinical stage. A key result of this study is that CSF p-tau231 is already significantly increased before overt Aβ pathology. CSF p-tau231 was significantly associated with Aβ PET retention in brain areas that are typically affected early in the AD, such as the medial orbitofrontal, precuneus and posterior cingulate cortices in cognitively unimpaired subjects[33]. ’’(lines 198-207 page 5)

2) Figure 1: misleading. First step in clinical practice is NRL-NPS assessment for MCI diagnosis, later AD diagnosis must be done using core biomarkers and checking for comorbidities (no diagnosis of MCI-AD is done without biomarkers); soon (in some places they have already started) for first screening they are using plasma biomarkers (ptau217 mainly) and also for monitoring. EEG and fMRI don't seem to be at the same level of other biomarker for monitoring. Explain APOE status could influence inclusion or not in case of anti-amyloid treatments.  

OUR RESPONSE: This is a figure that summarized the biomarkers that should be considered in the different phases of the AD clinical trials such as the Identification  and stratification of participants based on their genetic status ( if they are carriers  of  mutation such as genetic factors e.g.presenilins  PS1, PS2, APP, APOE4 ) , cognitive status as prodromal stages using plasma Aβ42/Aβ40, ptau assays or combined with APOE genotype

MRI, EEG and fMRI  are not at the same strong  level of evidence  as the other biomarker for monitoring, but they are often used in clinical trials.

APOE genotype is an inborn risk indicator rather than a biomarker for Aβ pathology (the CSF Aβ tests work independently of APOE genotype to detect cerebral Aβ pathology)

3) AT(N) biomarkers are NOT neurodegenerative biomarkers. Only (N) is indicative of neurodegeneration, and is the less relevant of the three categories. Explain how AT(N) biomarkers are NOT only relevant for atypical but also typical clinical syndromes for its early diagnosis.  

OUR RESPONSE: ‘’ During the last decade, the 3 “established” or “classical” cerebrospinal fluid (CSF) biomarkers for AD have been incorporated in diagnostic criteria/guidelines [1,3] and classification systems (ATN) [22]: The ATN research framework, proposed in 2011 and updated in 2018 by the NIA-AA, proposes to use biomarkers (namely amyloid (A), tau (T), and neurodegeneration (N)) to categorize individuals with an AD diagnosis. The framework was conceptualized for a biological construct of AD, not clinical symptoms of AD pathology. This ATN research framework can utilize CSF biomarkers where (a) the ratio of the two Amyloid-β Aβ peptides (CSF Aβ42/40) is a measure for A Amyloid-β peptide with 42 amino acids (Aβ42) which is decreased in AD, is considered as a marker of amyloid plaque pathology [23], (b) tau phosphorylated at threonine 181 (p-Tau 181) is a measure for T tau protein phosphorylated to a threonine residue at position 181 (τP-181) which is increased in AD, is considered as a marker of tangle formation [24] and (c) total tau protein (τT) is a measure for N which is increased in AD is a non-specific marker of neuronal and/or axonal degeneration[25]. The Aβ42/Aβ40 ratio may be preferred to Aβ42 alone since it seems to perform diagnostically better than the latter [26]. Plasma Aβ42/Aβ40 levels are fully changed already during the pre‐symptomatic disease stages; this is the reason this biomarker, like CSF Aβ42/Aβ40, can identify Aβ pathology in cognitively unimpaired people with accuracies as high as those observed in cognitively impaired individuals [27]. P-tau217 is considered to be the most robust among p-tau markers (p-tau181, p-tau 231, p-tau205) . In the diagnosis of AD, the performance of CSF p-tau217 is better than that of p-tau181 (area under the receiver operator characteristic curve (AUC), 0.943 vs 0.914, P = 0.026)[28] . At the same time, CSF p-tau217 can distinguish AD from other neurodegenerative diseases with dementia, and the accuracy is superior to that p181. Both plasma p‐tau181 and p‐tau217 have been shown to accurately predict future cognitive decline in patients with MCI, and conversion to AD dementia in the subsequent 2 to 6 years [29,30]. However, p-tau217 rises in the asymptomatic phase and changes with the progression of AD, allowing prediction and early diagnosis of AD, while higher p-tau217 levels suggest a faster cognitive decline [31]. Regarding the above advantages, p-tau217 is an appropriate biomarker with respect to the T in the peripheral A-T-N-X framework.. Importantly, plasma p‐tau231 might be changing slightly earlier than the other p‐tau markers[32]. CSF p-tau217 had the highest fold-change increases in symptomatic stages of the disease, while CSF p-tau231 better captured the earliest Aβ changes in the preclinical stage. A key result of this study is that CSF p-tau231 is already significantly increased before overt Aβ pathology. CSF p-tau231 was significantly associated with Aβ PET retention in brain areas that are typically affected early in the AD, such as the medial orbitofrontal, precuneus and posterior cingulate cortices in cognitively unimpaired subjects[33]. ’’(lines 164-207 page 5)

4) Overview fluid biomarkers: in the text or directly in the table 1  explain amount of changes in biomarkers and clinical/neurophsychological level for each drug and its potential relevance both from clinical and biological perspective.  

OUR RESPONSE: The table 1 was update according the reviewers suggestions with column entitled Biomarker changes, Clincal/neurophsychological outcomes , while the column ptau effect was incorporated in the column of the biomarkers changes. We also added a new column showing the Potential relevance both from clinical and biological perspective

5) Genetic biomarkers: explain better how APOE status could influence the capacity of inclusion in anti-amyloid therapies, also explain the newly suggested APOE4 related semidominant concept (APOE E4 homozygous as semidominant AD, Fortea et al 2024).

OUR RESPONSE: ‘’APOE genotype is an inborn risk indicator rather than a biomarker for Aβ pathology (the CSF Aβ tests work independently of APOE genotype to detect cerebral Aβ pathology). Knowledge of APOE genotype has, however, earned  enhanced clinical importance in the context of anti-Aβ immunotherapy. A recent study by Fortea et al.[19] suggested that APOE4 homozygotes represent a genetic form of AD : nearly full penetrance, predictability of symptom onset, and predictable sequence of biomarker changes. The risk of amyloid-related imaging abnormalities (ARIAs) is substantially greater in APOE ε4 homozygotes than in heterozygotes and non-carriers . Consequently, screening for APOE is recommended in the FDA label for lecanemab, and counseling around risk is recommended for homozygotes. APOE4 status must be recognized as a crucial parameter in clinical trial design, patient recruitment and data analysis, with AD risk across age, sex, race and ethnicity (stronger risk for East Asians vs Hispanics) for establishing the personalized AD therapy[20].’’ (lines 139-153  page 4)

6) The most innovative part of the review related with -omic strategies, those that clearly could be more related with personalized medicine are the one that are less explained during the manuscript. They are really the sections that could add extra value, together with more critical lecture of the literature, explaining how this advances should be incorporated to clinical practice (in a summary table) and also research gaps in the field and future steps that must  be done in order to try to solve them.  

OUR RESPONSE: ‘’There are generally three different stages in the development of new biomarkers: the discovery phase (i.e., screening), the verification phase, and the validation phase. Mass spectrometry (MS)-based proteomics technology is a novel growing body of  AD biomarker research in the establishment of AD precision medicine[94]. During the last 10 years, apart from the gel-based techniques (e.g., 2D-PAGE and 2D-DIGE), gel-free techniques (e.g., stable isotope labeling or using label-free methods) have been dominating the field of MS-based quantitation in proteomics[94]. iTRAQ with multidimensional liquid chromatography and tandem mass spectrometry has been used to reveal many candidate proteins as potential biomarkers of MCI or AD [95,96]. These proteins were found involved in various biological processes and pathways, such as Aβ metabolism, inflammatory and immune response, and oxidative stress, which have previously been reported to be linked with AD, supporting the existing theories of AD pathophysiology. Furthermore, new technologies such as SWATH-MS will also be applied to further enhance probability of AD biomarkers. SWATH-MS is a specific further variant of data-independent acquisition (DIA) methods and is emerging as a technology that combines deep proteome coverage capabilities with quantitative consistency and accuracy [97]. Apart from quantitative proteomics, the development of assays to quantify particular post-translational modification of proteins such as Western blotting with two-dimensional gel electrophoresis (2D-Oxyblot) identified the specifically carbonylated proteins in the hippocampi and serum of triple transgenic mouse model of AD (3 × Tg-AD) at the early age [98,99]. This finding suggests that oxidative stress is an early event in AD progression, and these oxidized proteins in the serums may provide potential biomarkers of AD at the early stage. In the same lines, increased levels of serum protein carbonylation were found in MCI subjects [100]. Together, the proteomic approach is comparatively new and more advanced for biomarker analysis of proteins and provides a complementary way to obtain such a comprehensive data.

  Multiple reaction monitoring (MRM), also known as selected reaction monitoring, is a targeted mass spectrometry approach to protein quantitation and is emerging to bridge the gap between biomarker discovery and clinical validation . Highly multiplexed MRM assays are readily configured and enable simultaneous verification of large numbers of candidates facilitating the development of biomarker panels which can increase specificity [101]. MRM can enhance the lower detection limit for peptides due to its ability to rapidly and continuously monitor exclusively for the specific ions of interest. MRM analysis combine with stable isotope also offers multiplexing capability and increases the reliability of quantification [101]. As AD is a multifactorial disease, a panel of proteins is more suitable as biomarker for AD. Thus, MRM is a valuable tool to verify biomarker candidates for AD and possible future practical applications. Several studies have emerged using MRM to identify CSF-based protein biomarkers of AD [102]. In addition to MRM, parallel reaction monitoring (PRM) technique has also been used to evaluate biomarker candidates for AD [103]. PRM is related to the SRM approach but has the advantage of acquiring full fragment spectra instead of a choice of preselected fragments; interfering signals are avoided, whereas quantitation and high sensitivity are conserved[104] . In this way, other biochemical pathways and proteins which are not directly correlated to Aβ accumulation could be monitored, such as synaptic function, secretory vesicle function, and in the innate immune system.

  Due to wide dynamic range and low abundance of Aβ peptides, the most common experimental procedure to quantitate Aβ peptides in CSF or blood requires a sample preparation step before MS analysis. Many methods are currently available to purify/concentrate the Aβ peptides, such as solid-phase extraction (SPE), immunoprecipitation (IP), size exclusion, ultrafiltration and liquid-liquid extraction, immunodepletion, etc.[104]. By using IP coupled with SRM-MS method, a recent publication reported that plasma Aβ42 concentration correlated with the CSF Aβ42/Aβ40 ratio and had good accuracy for predicting the sensitivity and specificity of elevated brain Aβ . Similarly, the amyloid-β precursor protein (APP) 669–711/Aβ42 and Aβ40/42 ratios and their composites, for AD diagnosis with an accuracy of 90% in predicting Aβ brain burden at an individual level, as confirmed with PET imaging [105]. Of note, in normal APP and Aβ metabolism, Aβ is most likely regulated by amyloid-degrading enzymes [103]. Different lengths of Aβ peptides exist in vivo, depending on different degradation pathways of APP [106]. Notably, these approaches can not only give a more accurate quantification of Aβ peptides in blood or CSF but also can be used to detect various Aβ species, which are beneficial to screen candidate biomarker for AD. For example, using the high selectivity of anti-Aβ antibodies in combination with mass spectrometry to determine the molecular mass with high accuracy, Vigo-Pelfrey et al. demonstrated the complex nature of Aβ peptides in the CSF and reported several different N- and C-terminal variants of Aβ[40] . IP-MS method has also been used to measure the protein levels in the CSF; using this method, a marked increase in the CSF levels of both synaptosomal-associated protein 25 (SNAP-25) and synaptotagmin-1 (SYT1) was found in AD dementia and prodromal AD cases [107]. Interestingly, the levels of both SNAP-25 and SYT1 are reduced in cortical areas in the AD brain [90], thus suggesting that a set of synaptic proteins covering different components of the synaptic unit may be valuable tools in clinical studies on the relevance of synaptic dysfunction and degeneration in AD pathogenesis. This strategy is advantageous for detecting low abundance proteins, especially from CNS, or various Aβ peptides as a target biomarker of AD potentially for precision AD diagnosis and therapy.’’ (lines 379-383 page 6, lines 384-423 page 7, lines 424-435 page 8)

We add more critical lecture of the literature with  research gaps in the field and future steps that must  be done to  incorporated to clinical practice as following (see after Discussion part):

5.Research Gaps

1.Lack of certified biofluid reference methods and materials (except for cerebrospinal fluid [CSF] amyloid beta [Aβ]42, where these are available).

2.The methods of RNA and exosome isolation, and downstream miRNA detection, quantification and normalization methods varied between studies such as enzyme-linked immunosorbent assays (ELISA), Western blotting, and mass spectrometry. S, leading to conflicting results.

  1. There is a paucity of comprehensive biofluids analyses assessing CSF and blood levels of multiple inflammatory markers along with Core 1 and 2 biomarkers.
  2. In order to enrich cohorts for maximized therapeutic effects in clinical trials, knowledge of the predictive/prognostic value of omic profiles in relation to clinical trajectories is crucial.

5.Despite the efforts, PET, CSF, and blood biomarkers remain less sensitive compared with neuropathologic examination for detection of early/mild AD neuropathologic change (ADNPC). Disease staging by PET (or fluid biomarkers) is not equivalent to neuropathological staging, for example, tau PET ligand uptake in different Braak areas is not equivalent to Braak neuropathological staging. While the sensitivity limits of biomarkers could be appraised as a disadvantage, they could also be appraised as a strength because abnormal Core 1 biomarkers indicate that ADNPC more generally rather than just neuritic plaques alone is very likely present.

  1. Thoroughly studied biomarkers are not available for all relevant diseases; there is a high uncertainty of other copathologies in addition to AD in any individual, or what the proportional diseasespecific burden is among various pathologic entities.
  2. The proportion of the cognitive deficit observed in a single patient that is attributable to AD versus other neuropathologic pathologies is difficult to be quantified. Only probabilistic rates can be calculated based on combinations of biomarker results and clinical evaluation.
  3. Future Steps
  4. Future protocols of clinical trials should rigorously include more representative cohorts. True epidemiolocal and real-world data studies of biomarker properties in representative groups are crucial to determine relationships that are valid at the population level. Better understanding of the longitudinal intra‐individual biological and disease‐associated variability and potential impact of clinical confounders and biological factors, including race and ethnicity, peripheral neuropathies and other neurologic diseases, BMI, and kidney disease and the relative effects on the clinical performance of plasma Aβ42/Aβ40, p-tau, NfL, and GFAP in large cohorts
  5. Loger clinical trials are needed to show the lowering rate of brain volume loss as a result of the amyloid plaque removal.

3.An international consensus of  standard biofluid assays, tau PET quantification methods, and cutpoints is warranted. As in other diseases, the exact thresholds for abnormality may evolve over time as additional data inform prognostic value.

  1. Advanced knowledge of various posttranslational modifications of tau may enhance fluid-based biological staging.
  2. With improved understanding of the role of immune/inflammatory processes, microglia, and astrocyte biology in AD pathogenesis, we foresee a more notable role for I biomarkers in biological characterization and prognosis, especially if brain-specific modifications can be revealed in blood.
  3. Keeping in mind that clinical trials targeting mechanisms other than anti-Aβ immunotherapy, the effects of these interventions on biomarkers and clinical outcomes should be included in future diagnostic AD criteria.

Reviewer 2 Report

Comments and Suggestions for Authors

Dear authors,

Thank you very much for the submission.

This review well established the biomarkers for early AD prediction and detection from multiple aspects, including but not limited to genetics, neuroimaging, and fluids. These demonstrations all well provided promising precise AD detection from case to case which will provide some hints for further Alzheimer's precision therapy based on the biomarkers in individuals.

However, there are still some issues that need to be further addressed.

1: In general, this review mainly discussed the biomarkers for AD prediction and detection, however, not really mention much about how the AD therapy will be performed and designed based on the biomarkers in each individual and how to provide the precise therapy. Please add more information based on that.

2: There are plenty of typo, spacing, and font issues throughout this article, for instance, line 32, 34,75, 97, 220, 236, 236, 239 (should be NED instead of NDE), 269, 286, and line 304. Also, the font of the title from 3.4, 3.5, 3.6 are different from 3.1, 3.2, 3.3 title. Please check carefully and revise accordingly.

Author Response

REVIEWER 2

Thank you very much for the submission.

This review well established the biomarkers for early AD prediction and detection from multiple aspects, including but not limited to genetics, neuroimaging, and fluids. These demonstrations all well provided promising precise AD detection from case to case which will provide some hints for further Alzheimer's precision therapy based on the biomarkers in individuals.

However, there are still some issues that need to be further addressed.

1: In general, this review mainly discussed the biomarkers for AD prediction and detection, however, not really mention much about how the AD therapy will be performed and designed based on the biomarkers in each individual and how to provide the precise therapy. Please add more information based on that.

OUR RESPONSE: We dedicated an important part of the biomarkers  in this review because biomarkers greatly contribute to correctly assessing a patient's status in personalized medicine, as they provide invaluable insight into individual biological characteristics and the process of disease .Amyloid biomarkers establish the presence of the target pathology for anti-amyloid trials. Amyloid biomarkers show that the patient has AD and not some other unknown pathology that could create a neuronal environment with an unknown therapeutic response to anti-amyloid treatments.

In this regard, ‘’Τhree potential molecular biomarkers, namely monoacylglycerol lipase (Mgll), apolipoprotein E4 (APOE4) and the phosphatidylinositol 3-kinase (PIK3)/protein kinase (AKT)/glycogen synthase kinase-3β (GSK-3β) signaling pathway, as prime candidates for targeted potential personalized therapy[50]. Importantly, using a transgenic mouse model CbpS436A, where the aPKC-CBP pathway is deficient, monoacylglycerol lipase (Mgll) gene expression was directly repressed upon activation of the aPKC-CBP pathway with metformin treatment. Coincidently, Mgll levels were abnormally upregulated in 3xTg mice during the aging process and that metformin was able to reactivate the impaired aPKC-CBP pathway to repress Mgll expression, thus rescuing both hippocampal neuronal differentiation and spatial memory deficits in 3xTg mice. In this regard, Mgll is a perfect candidate biomarker to identify prospective patients in the early stages of AD that are best suited to receive metformin as a treatment against the disease. Anti-APOE4 immunotherapy involving the use of antibodies to target and neutralize APOE4 protein as a potential precision therapeutic strategy of AD[51]. Another strategy is the development of small molecules to block APOE4 domain interactions to counteract its pathological effects. Since APOE4 is susceptible to degradation by neuronal proteases that yield neurotoxic fragments, the identification of these proteases and the development of inhibitors against them will also help against APOE4 toxicity. In addition, since APOE4 lowers the levels of APOE2 receptors, one possible therapeutic approach could be to increase the expression of APOE2 receptors to promote its “protective” effect versus the “toxic” effect of APOE4 [51].

    When the PI3K/AKT signaling pathway is dysfunctional, such as due to insulin resistance, it increases GSK-3β activity and leads to Tau hyperphosphorylation, predisposing to AD[50]. This regulation of the PI3K/AKT/GSK-3β pathway appears to be crucial for AD pathogenesis [52](Kitagishi et al., 2014). Thus, targeted therapy to reduce GSK-3β activity has become a promising approach to treat AD[53]. Lithium, a mood stabilizer used in patients suffering from mood disorders, is now known to both, directly and indirectly, inhibit GSK-3β activity. While lithium administration successfully reduced the neuropathology and cognitive deficits in rats that received intra-hippocampal injections of Aβ, in rats overexpressing GSK-3β, as well as, in several murine models overexpressing human amyloid precursor protein, it failed to show promise in other murine models of AD[53]. Thus, lithium treatment may only be beneficial against AD-associated cognitive deficits and neuropathology in patients exhibiting abnormal GSK-3β activity [54]. While no GSK-3β-specific neuro-radiotracer has been approved for use in humans yet, one compound has shown great promise in the primate brain.’’ (lines 269-280 page6, lines 281-310 page 7)

2: There are plenty of typo, spacing, and font issues throughout this article, for instance, line 32, 34,75, 97, 220, 236, 236, 239 (should be NED instead of NDE), 269, 286, and line 304. Also, the font of the title from 3.4, 3.5, 3.6 are different from 3.1, 3.2, 3.3 title. Please check carefully and revise accordingly.

OUR RESPONSE: We correct all the issues such as typo, spacing, and font issues throughout this article, in  line 32, 34,75, 97, 220, 236, 236, 239 (NED instead of NDE), 269, 286, and line 304. Also, the font of the title from 3.4, 3.5, 3.6 was corrected.

Round 2

Reviewer 1 Report

Comments and Suggestions for Authors

Thank you for modifying significantly your manuscript. I think that you have considerable improved the quality of your work, reorganizing introduction section, adding extra information useful for clinicians and a more critical interpretation of the review done by including research gap and future research ideas sections.

However, I should suggest the next modifications to the authors previous to consider my potential endorsement of this publication:

INTRODUCTION SECTION:

Please modify this sentence: "Considering their time of onset, plasma 55 p-tau217and p-tau231 have been suggested as biomarkers of Aβ plaques, but this is not  conceptually correct because of the coexistence of tau fragments"
** I don't think it is appropiate to say that it is not conceptually correct. A+ also could be established by AB42/ptau ratio in the current and past criteria. You could say that you may prefer to not combine A+ and T+ biomarkers to define A+, but not say that it is not correct.

Add information about how AD is diagnosed in MCI stage: NRL-NPS assessment (not diagnosis in current clinical practice in preclinical stages) and A+T+. V and other biomarkers are to check copathology. Please rewrite this part. You have improved significantly but it is not clear enough.

FIGURE 2:

I don't understand if really summarize extra useful information.

I would create another table to address the same objectives but with other perspective: 1) Selection useful biomarkers (could be demographic data based such as CAIDE dementia risk score, NPS parameters -obtained using remote monitoring for istance-, plasma AT biomarkers could be useful to detect preclinical cases and not only genetic risk factors mentioned; 2) Diagnosis of AD: 1) First explain if the focus is going to be symptomatic AD and in that case add first NPS assessment, test that must be done before considering CSF, PET or plasma biomarker study (explain A+T+ is mandatory and exclusion of comorbidities should be done); 3) Monitoring or response biomarkers: should include obviously NPS outcomes.

CLINICAL TRIALS WITH ANTIAMYLOID DRUGS:

I would not start this section with the next sentence: "The investigation of donanemab (target class: amyloid β) in the TRAILBLAZER-ALZ research was unable to find significant alterations in plasma Aβ42/40 ratio levels [43]. "

** First explain donanemab mechanism and later the positive results (clinical and biomarkers based) and later (obviously should be mentioned) the not so robust results.

NEUROIMAGING BIOMARKERS:

Summarize in a table current utility in clinical practice and trials of available MRI and PET biomarkers and their utility in research context (not only structural MRI, also functional, mean diffusivity studies to check inflammation, metabolism variation studies using MRI.... and also refer to FDG-PET).

GENETIC BIOMAKERS.

Mention that you are refering sporadic AD, and explain the concept of genetic AD including DS population, in the last one the presence of APOE4 could also modify the age of onset symptoms and biomarkers positivity moment. 

INSTEAD OF MICRORNA SECTION, HAVE YOU CONSIDERED TO CALL IT EPIGENOMICS

You could also information about other mechanisms such us DNA methylation, quite studied in AD and with potential utility to improve the capacity of diagnosis of AD. Please check the posibility to add this information.

IF YOU SUGGEST EEG or other neurophysiological tools that could sleep assessment please add evidence refering to their potential utility, not just mention them in a table at the first part of the manuscript

Author Response

Thank you for modifying significantly your manuscript. I think that you have considerable improved the quality of your work, reorganizing introduction section, adding extra information useful for clinicians and a more critical interpretation of the review done by including research gap and future research ideas sections.

However, I should suggest the next modifications to the authors previous to consider my potential endorsement of this publication:

INTRODUCTION SECTION:

Please modify this sentence: "Considering their time of onset, plasma 55 p-tau217and p-tau231 have been suggested as biomarkers of Aβ plaques, but this is not  conceptually correct because of the coexistence of tau fragments"
** I don't think it is appropiate to say that it is not conceptually correct. A+ also could be established by AB42/ptau ratio in the current and past criteria. You could say that you may prefer to not combine A+ and T+ biomarkers to define A+, but not say that it is not correct.

OUR RESPONSE: We corrected as: ‘’Considering their time of onset, plasma 55 p-tau217and p-tau231 have been suggested as biomarkers of Aβ plaques, but it is preferable  not combine A+ and T+ biomarkers to define A+ because of the coexistence of tau fragments’’ (lines page)

Add information about how AD is diagnosed in MCI stage: NRL-NPS assessment (not diagnosis in current clinical practice in preclinical stages) and A+T+. V and other biomarkers are to check copathology. Please rewrite this part. You have improved significantly but it is not clear enough.

OUR RESPONSE: ‘’According  the International Working Group on Mild Cognitive Impairment (IWGMCI), different cognitive phenotypes could arise from different cognitive domains being affected independently of memory, and that subjective complaints were no longer necessary. These new criteria, which provide etiological and prognostic characterizations of clinical utility, include the distinction between amnestic and non-amnestic MCI subtypes as well as whether cognitive impairment is restricted to a single domain or numerous domains. The IWGMCI agreement said that biomarkers could be useful in clarifying clinical progression and offered a flexible framework for MCI diagnosis. It would be beneficial to use therapy in the early stages of the disease, when these interventions may be more successful, in order to anticipate the progression of these MCI patients towards dementia.’’ (lines 82-97 page2)

FIGURE 2:

I don't understand if really summarize extra useful information.

I would create another table to address the same objectives but with other perspective: 1) Selection useful biomarkers (could be demographic data based such as CAIDE dementia risk score, NPS parameters -obtained using remote monitoring for istance-, plasma AT biomarkers could be useful to detect preclinical cases and not only genetic risk factors mentioned; 2) Diagnosis of AD: 1) First explain if the focus is going to be symptomatic AD and in that case add first NPS assessment, test that must be done before considering CSF, PET or plasma biomarker study (explain A+T+ is mandatory and exclusion of comorbidities should be done); 3) Monitoring or response biomarkers: should include obviously NPS outcomes.

OUR RESPONSE:The FIGURE 2 was replaced by a table 1as reviewer suggested:

The A,T,N Research Framework incorporates biomarkers into the diagnosis process of AD and has applications in clinical trials and medication development. The FDA's staging approach for AD makes it easier to develop drugs for the predementia phases of the disease and incorporates biomarkers into it[43], as Table1 shows. Diagnostic biomarkers provide precise diagnosis and enable classification of a disease based on the existence or lack of a certain pathophysiological state. In order to maximize the establishment of a drug-placebo difference, predictive biomarkers can be utilized to enrich populations and indicate the development of the disease. Treatment response prediction is made easier with the use of predictive biomarkers. Pharmacodynamic or activity biomarkers indicate the occurrence of a biological reaction in the patient receiving the therapeutic intervention. Safety biomarkers (biochemical, MRI, EEG) are biomarkers for identifying unfavorable and unintended medication responses. (lines 237-251 page 6)

Table 1.  Role of biomarkers for each phase of AD drug development based on FDA guidelines

Biomarkers

Screening

Phase 1

Phase 2

Phase 3

Diagnostic biomarkers  of AD

Low CSF Aβ42 or CSF Aβ42/t-tau ratio or Aβ42/ptau ratio or positive amyloidPET

demographic data based such as CAIDE dementia risk score, ADAS-Cog

symptomatic AD

A+T+ is mandatory and exclusion of comorbidities should be done

Predictive biomarkers

tau PET used to determine whether AD patients are more likely to benefit from anti-tau treatments

Prognostic biomarkers:Sort people based on likelihood of illness or include more patients in trials

Tau PET to determine which AD patients are most likely to experience cognitive decline more quickly
ApoE-4 carriers in immunotherapy studies as a prognostic marker for ARIA

Tau PET to determine which AD patients are most likely to experience cognitive decline more quickly
ApoE-4 carriers in immunotherapy studies as a prognostic marker for ARIA

Pharmacodynamic biomarkers:

i)Target engagement

ii)Disease modification (atrophy on MRI, hypometabolism on FDG PET, or increases in total tau in the CSF)

Phase 2's essential result for moving on to Phase 3

Essential outcome for the intervention to be classified as a DMT

Safety biomarkers

In immunotherapy regimens, liver function and other laboratory tests, an ECG, and an MRI are used to check for ARIA.

In immunotherapy regimens, liver function and other laboratory tests, an ECG, and an MRI are used to check for ARIA.

Liver function and other laboratory tests, ECG, MRI to monitor for ARIA in immunotherapy programs

ApoE apolipoprotein E, ARIA amyloid-related imaging abnormalities, CSF cerebrospinal fluid, FDG fluorodeoxyglucose, MRI magnetic resonance imaging, PET positron-emission tomography

CLINICAL TRIALS WITH ANTIAMYLOID DRUGS:

I would not start this section with the next sentence: "The investigation of donanemab (target class: amyloid β) in the TRAILBLAZER-ALZ research was unable to find significant alterations in plasma Aβ42/40 ratio levels [43]. "

** First explain donanemab mechanism and later the positive results (clinical and biomarkers based) and later (obviously should be mentioned) the not so robust results.

 OUR RESPONSE:   ‘’Donanemab is an immunoglobulin G1 monoclonal antibody that targets the insoluble, shortened form of β-amyloid that is only found in brain amyloid plaques and has been changed. Donanemab binds to the β-amyloid's N-terminally shortened version, facilitating the phagocytosis of microglia that removes plaque. Following donanemab, there was a substantial correlation found between the Centiloid percent change in amyloid and changes in plasma pTau217 and glial fibrillary acidic protein. Furthermore, there was a strong correlation between the plasma levels of pTau217 and glial fibrillary acidic protein both before and after treatment. The investigation of donanemab (target class: amyloid β) in the TRAILBLAZER-ALZ research was unable to find significant alterations in plasma Aβ42/40 ratio levels [43]. " (lines 263-271 page7)

NEUROIMAGING BIOMARKERS:

Summarize in a table current utility in clinical practice and trials of available MRI and PET biomarkers and their utility in research context (not only structural MRI, also functional, mean diffusivity studies to check inflammation, metabolism variation studies using MRI.... and also refer to FDG-PET).

 OUR RESPONSE: We summarized the utility in research context, clinical practice and trials of neuroimaging AD biomarkers, as reviewer suggested:

Table3. Utilityof neuroimaging AD biomarkers in research context, clinical practice and trials

Type of neuroimaging biomarker

Utilityt in research context

Utility in clinical practice and trials

Structural MRI

 Atrophy of the hippocampus or the surrounding medial temporal lobe regions

DWI

more indicative of early progressive cognitive change (

Functional MRI

 less connection between the medial temporal regions and the posterior cingulate cortex. 

not recommended for routine clinical usage (high cost, limited spatial resolution)

FDG PET

reflective of synaptic activity and neuronal activating

deficits in regional cerebral blood flowpredicting conversion to AD in people with MCI

elevated microglial activity as an inflammatory marker to monitor the anti-inflammatory effects of AD treatments

Amyloid PET

recognizing the intermediate-high neuropathologic alteration of AD

The retention time of PiB indicates the change of MCI to AD.

Tau PET

measures the fibrillar deposited form of the tau protein  to monitor in anti-tau trials

AD: Alzheimer’s disease, DWI : Diffusion-weighted imaging,  FDG-PET: 18F-fluorodeoxyglucose-PET, MRI: magnetic resonance imaging

GENETIC BIOMAKERS.

Mention that you are refering sporadic AD, and explain the concept of genetic AD including DS population, in the last one the presence of APOE4 could also modify the age of onset symptoms and biomarkers positivity moment. 

OUR RESPONSE:  Regarding the concept of genetic AD:’’ The general population is frequently (95%) affected by sporadic AD, which manifests as late-onset AD (LOAD) in people over 65. Age, female sex, traumatic brain injury, depression, environmental pollution, physical inactivity, social isolation, low academic level, metabolic syndrome, and genetic susceptibility—primarily mutations in the ε4 allele of apolipoprotein E (APOE, 19q13.32) —are the main risk factors of sporadic AD[61]. Heritability of the condition can reach 60–80% .The familial form of genetic AD is autosomal dominant, early onset (EOAD) in people under 65 (affecting 1 to 5% of cases), and typified by mutations in particular genes, including presenilin 1 (PSEN1, 14q24.2), which has been found to be altered in up to 70% of cases of familial AD; presenilin 2 (PSEN2, 1q42.13); and the Amyloid precursor protein gene (APP, 21q21.3).’’ (lines 366-373 page5)

Regarding the DS population: ‘’The genetic basis for amyloid precursor protein profusion in Trisomy 21, also known as Down syndrome (DS), is EOAD. Because to the overabundance of Aβ and the amyloid precursor protein, by the mid-40s, all DS patients have enough ADNPC to meet the neuropathological criteria for an AD diagnosis[67]. The same level of genetic penetrance as in autosomal dominant AD (ADAD) is consistent with the age at onset and mortality in DS. With a mid-50s typical age of onset for clinical symptoms, the lifetime probability of dementia is 95% in DS[68]. Increased levels of peripheral proteins including Aβ40, Aβ42, MMP-1, 3 & 9, proNFG, and inflammatory mediators like IFN-γ, TNF-α, IL-6, IL-10, and IL-1 were among the changes in plasma biomarkers found in DS[69] . There is, however, always some degree of doubt regarding the precise timing of these changes as well as whether the altered biomarkers are caused by inherited AD or DS. Notably, Aβ1-42/1-40 levels in cerebrospinal fluid decreased, hippocampi shortened, plaque burdens increased, cortical metabolism slowed, and plasma phospho-tau181 levels rose sooner in individuals with Down's syndrome and ApoE4[20,70]. There were no differences in CSF p-tau181, total tau, or both fluid NfL levels [70].’’ (lines 383-388 page5, lines 387-391 page 6)

INSTEAD OF MICRORNA SECTION, HAVE YOU CONSIDERED TO CALL IT EPIGENOMICS OUR RESPONSE:OK We renamed it in EPIGENOMICS

You could also information about other mechanisms such us DNA methylation, quite studied in AD and with potential utility to improve the capacity of diagnosis of AD. Please check the posibility to add this information.

 OUR RESPONSE: Thank you for this valuable suggestion. We add the epigenetics mechanisms such us DNA methylations in the Epigenomics Section (prior named as microRNAs) as:’’ 3.6. MicroRNA (miRNA) Epigenomics

 Any process via which the environment can modify a phenotype without changing the genotype is known as epigenetic alterations, and they may necessitate a signaling cascade from the production of transcription factors. There are currently over twenty recognized epigenetic mechanisms, such as DNA methylations, genomic imprinting, noncoding RNAs (ncRNAs), post-translational modifications of histones (PTM-Hs) that alter gene expression by activating or repressing it, and a variety of confounding variables associated with changes in the environment.

  Dysregulation of miRNA,  small ncRNAs of 20–22 nucleotides in length, which regulate the half gene expression post-transcriptionally by binding to the 3' untranslated region (UTR) of target mRNAs and are implicated in various neurodegenerative disorders, including AD, where miRNAs can modulate the expression of genes involved in amyloid-beta metabolism, tau phosphorylation, neuroinflammation, and synaptic dysfunction [86]. Through the sequential activity of cleavage enzymes BACE1 and γ-secretase, miRNAs can modify the processing of amyloidogenic APP into neurotoxic Aβ-42/40 and p-tau aggregates by modulating the target genes. Tauopathy and the development of amyloid plaques are encouraged when the CAMK4 gene, which controls synaptic activities in neuronal cells, is inhibited by microRNAs. Likewise, disruption of the Dicer/Drosha complex results in the termination of miRNA production and is linked, albeit indirectly, to the deregulation of DNMT enzymes and consequently, DNA methylation. In AD brains, the ADAM10 gene is implicated in APP processing and Aβ-amyloidosis; it is overexpressed due to particular miRNA molecules inhibiting the gene….”(lines 532-546 page10, lines 547-549 page 11)

IF YOU SUGGEST EEG or other neurophysiological tools that could sleep assessment please add evidence refering to their potential utility, not just mention them in a table at the first part of the manuscript OUR RESPONSE: we eliminated it just not to be confused
